# Towards One-for-All Anomaly Detection for Tabular Data

**Shiyuan Li**[1]  **Yixin Liu**[1]  **Yu Zheng**[1]  **Xiaofeng Cao**[2]  **Shirui Pan**[1]  **Heng Tao Shen**[2]

## Abstract

Tabular anomaly detection (TAD) aims to identify samples that deviate from the majority in tabular data and is critical in many real-world applications. However, existing methods follow a "one model for one dataset (OFO)" paradigm, which relies on dataset-specific training and thus incurs high computational cost and yields limited generalization to unseen domains. To address these limitations, we propose OFA-TAD, a generalist one-for-all (OFA) TAD framework that only requires one-time training on multiple source datasets and can generalize to unseen datasets from diverse domains on-the-fly. To realize one-for-all tabular anomaly detection, OFA-TAD extracts neighbor-distance patterns as transferable cues, and introduces multi-view neighbor-distance representations from multiple transformation-induced metric spaces to mitigate the transformation sensitivity of distance profiles. To adaptively combine multi-view distance evidence, a Mixture-of-Experts (MoE) scoring network is employed for view-specific anomaly scoring and entropy-regularized gated fusion, with a multi-strategy anomaly synthesis mechanism to support training under the one-class constraint. Extensive experiments on 34 datasets from 14 domains demonstrate that OFA-TAD achieves superior anomaly detection performance and strong cross-domain generalizability under the strict OFA setting. The source code is available at https://github.com/Shiy-Li/OFA-TAD.

## 1. Introduction

Tabular data anomaly detection (TAD) aims to identify anomalies that significantly deviate from the majority of normal samples in tabular data (Han et al., 2022). As the

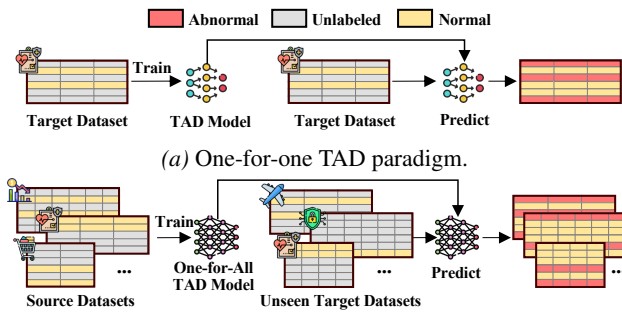

*(a) One-for-one TAD paradigm.*

*(b) One-for-all TAD paradigm.*

*Figure 1.* The difference between OFO and OFA paradigm.

volume of tabular data continues to grow, TAD is crucial for various scientific and industrial processes, including medical disease diagnosis (Fernando et al., 2021), cybersecurity (Ahmad et al., 2021), and financial fraud detection (Al-Hashedi & Magalingam, 2021). Conventional TAD methods, such as Isolation Forest (IForest) and Local Outlier Factor (LOF), rely on handcrafted heuristics and often struggle to capture complex non-linear relationships (Liu et al., 2008). Recently, deep learning-based approaches have acquired improved performance by learning in a one-class setting with only normal data only, using various unsupervised objectives to model intrinsic irregularities of the data distribution and identify anomalous samples as deviations (Qiu et al., 2021; Yin et al., 2024; Ye et al., 2025a).

To date, most existing TAD methods adopt a "one model for one dataset (OFO)" paradigm, where a dedicated TAD model is built and trained for each dataset independently, as shown in Figure 1a. Despite their strong in-domain performance, this paradigm poses major barriers to practical real-world deployment: ❶ *Prohibitive training cost.* For every new domain or dataset, the OFO paradigm requires training a detector from scratch, and often incurs costly hyperparameter search or even architecture re-design to achieve satisfactory performance. This requirement leads to substantial computation and operational overhead and makes large-scale rollout expensive in real-world scenarios. ❷ *Poor generalizability.* OFO models often overfit to the source-data distribution and hence transfer poorly to unseen data. Hence, they struggle to generalize to new scenarios and cannot effectively handle distribution shifts, resulting in unreliable performance after deployment.

[1]Griffith University, Gold Coast, Australia [2]Tongji University, Shanghai, China. Correspondence to: Shirui Pan <s.pan@griffith.edu.au>.

*Proceedings of the 43rd International Conference on Machine Learning*, Seoul, South Korea. PMLR 306, 2026. Copyright 2026 by the author(s).

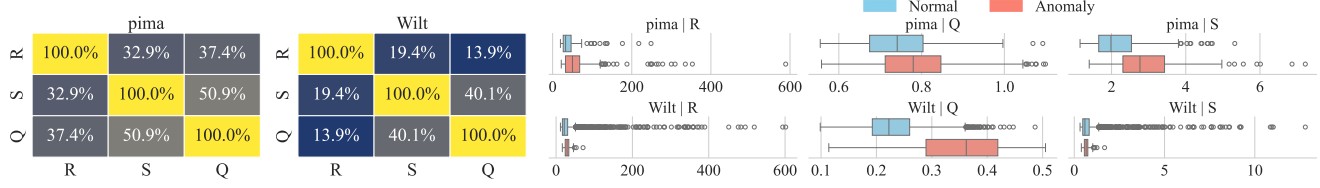

*(a)* The neighbor overlap between transformations.      *(b)* Distance sensitivity w.r.t. different transformations.

*Figure 2.* The distance varies in different transformations, where R: Raw, S: Standardized, and Q: Quantile.

A promising solution to these limitations is to embrace a "one model for all datasets (OFA)" paradigm, which aligns with the recent surge of foundation models and artificial general intelligence. As shown in Figure 1b, in the OFA paradigm, a single generalist TAD model is trained once on multiple-source datasets, and then it can be directly transferred to unseen datasets without any target-domain retraining or finetuning. Compared to its OFO counterpart, the OFA paradigm eliminates dataset-specific dependence by consolidating heterogeneous data patterns into a unified detector, and hence reduces computational redundancy while improving robustness and adaptability of the TAD model.

Despite the growing success of generalist OFA models in anomaly detection for other modalities (e.g., images (Zhu & Pang, 2024) and graphs (Zheng et al., 2022; Zhao et al., 2025; Liu et al., 2024; 2026; Qiao et al., 2025)), building an OFA tabular anomaly detector still remains under-explored. A primary obstacle is the "*semantic gap*": tabular data from diverse domains differ in dimensionality and feature semantics, and thus anomaly patterns are often domain-specific rather than universal. For example, medical anomalies may manifest as atypical physiological measurements (e.g., blood pressure and heart rate), whereas financial anomalies are reflected in irregular transaction behaviors (e.g., amounts and balances). As a result, a detector can easily rely on cues that work well in one domain but break when transferred to a new domain. This motivates the first challenge: ***Challenge 1 - How to discover transferable anomaly patterns that hold across domains?***

To address this challenge, we argue that transferable anomaly patterns should be grounded in universal neighborhood structures, which remain meaningful even when feature semantics vary across domains. At its core, anomaly detection aims to identify samples that are more isolated than normal points, i.e., those that lie unusually far from their local neighborhood (Goodge et al., 2022). In this case, we can capture this isolation by the "**neighbor distance**" profile (i.e., the Top-$k$ nearest-neighbor distance sequences) and use it as a semantics-agnostic representation to capture transferable anomaly patterns. For example, a patient record with abnormal vital signs or a fraudulent transaction with irregular spending behavior may both deviate from the bulk of normal samples; consequently, their Top-$k$ distance se-

quences often exhibit a pronounced elbow and a heavy tail, indicating the shared distance-level anomaly signature.

While the neighbor distance patterns are potentially transferable, the distance distributions can be highly sensitive to the feature transformations applied to the raw tabular data, which may undermine cross-domain generalization. Due to the heterogeneous scale and distribution of tabular features, feature transformations (e.g., normalization and standardization) are a common pre-processing step for TAD. Nevertheless, the selection of transformations can significantly change the neighborhood structure and distance distribution. As shown in Figure 2a, the Top-$k$ nearest-neighbor overlap of the same sample under different transformations can be remarkably low, indicating that transformations reshape the neighbor set and distance ranking. Moreover, Figure 2b shows that anomaly separability can vary drastically across transformations, and the optimal transformation differs considerably across datasets. Together, these results demonstrate pronounced *transformation sensitivity*: different datasets favor different transformations, and a fixed transformation may be insufficient to generalize to all unseen domains. While the OFO methods can leverage prior knowledge or trial-and-error to select a suitable dataset-specific transformation, in OFA settings, the transformations have to be determined automatically without access to target-domain supervision. This leads to the ***Challenge 2 - How can we construct robust distance representation under suitable transformations without prior knowledge?***

To address this challenge, we propose OFA-TAD, a multi-view distance-centric learning framework for OFA anomaly detection. Our theme is to treat neighbor-distance profiles under different transformations as complementary data views, and then use a Mixture-of-Experts (MoE) fusion mechanism to adaptively integrate them for robust anomaly scoring. Specifically, we first construct multiple metric spaces induced by different transformations, and for each sample, we extract a multi-view distance profile (Top-$k$ neighbor distance sequences) and encode it into cross-domain comparable representations via distribution normalization.

Then, we implement the MoE fusion with view-specific experts that use attention pooling to produce expert scores, while a learned gating network adaptively combines them

into the final anomaly score. Additionally, to handle the absence of true anomalies in the one-class setting, we synthesize diverse pseudo-anomalies via multi-strategy negative sampling, supporting the end-to-end optimization of OFA-TAD. This paper makes the following contributions:

- **Problem.** We, for the first time, propose a TAD paradigm shift from one-for-one to one-for-all, aiming to detect anomalies across various datasets with a single TAD model without dataset-specific fine-tuning.
- **Methodology.** We propose a novel one-for-all TAD model OFA-TAD, which can detect anomalies in new tabular datasets on-the-fly via multi-view distance encoding and MoE anomaly scoring.
- **Experiments.** We conduct extensive experiments to validate the detection capability and generalizability of OFA-TAD across 34 datasets from various domains.

## 2. Preliminary

In this section, we introduce the notations and problem definition. A review of related work is provided in Appendix A.

**Notations.** Let $\mathcal{D} = (\mathbf{X}, \mathbf{y})$ denote a tabular dataset, where $\mathbf{X} = [\mathbf{x}_1; \ldots; \mathbf{x}_n] \in \mathbb{R}^{n \times d}$ is the feature matrix and $\mathbf{y} \in \{0, 1\}^n$ is the anomaly label vector, with $y_i = 1$ indicating an anomaly and $y_i = 0$ indicating a normal sample. Following previous works, we study TAD in a one-class setting: the training set $\mathcal{D}_{train}$ contains only normal samples, while the test set $\mathcal{D}_{test}$ contains both normal and anomalous samples. We use $\mathbf{X}_{train}$ to denote the feature matrix of $\mathcal{D}_{train}$. The goal of TAD is to learn an anomaly scoring function $f : \mathbb{R}^d \to \mathbb{R}$ such that $f(\mathbf{x}_a) > f(\mathbf{x}_n)$ for anomalous samples $\mathbf{x}_a$ and normal samples $\mathbf{x}_n$.

**One-for-one TAD Problem.** In the conventional one-for-one setting ("one model for one dataset"), given a dataset $\mathcal{D} = \mathcal{D}_{train} \cup \mathcal{D}_{test}$, a detector $f$ is optimized on $\mathcal{D}_{train}$. After sufficient training, the learned model $f(\cdot)$ is directly applied to the $\mathcal{D}_{test}$ to output anomaly scores for each sample, where higher scores indicate higher abnormality.

**One-for-all TAD Problem.** This paper studies the One-for-All (OFA) setting, where a single detector is trained once and then deployed to multiple unseen target datasets. Let $\mathbb{D}_{all} = \{\mathcal{D}_j\}_{j=1}^N$ denote the dataset universe, where each dataset $\mathcal{D}_j = \mathcal{D}_{j,train} \cup \mathcal{D}_{j,test}$ follows the one-class protocol. $\mathbb{D}_{all}$ can be partitioned into source domains $\mathbb{D}_{source}$ and target domains $\mathbb{D}_{target}$ with $\mathbb{D}_{source} \cap \mathbb{D}_{target} = \emptyset$. The training sets of all source datasets form a unified cross-domain training pool $\mathcal{P}_{train} = \bigcup_{\mathcal{D}_j \in \mathbb{D}_{source}} \mathcal{D}_{j,train}$. A model $f(\cdot)$ is trained only once on $\mathcal{P}_{train}$; During evaluation, for each target dataset $\mathcal{D}_j \in \mathbb{D}_{target}$, $f(\cdot)$ directly predict anomaly scores on $\mathcal{D}_{j,test}$ by using $\mathcal{D}_{j,train}$ as domain context, without any retraining or dataset-specific tuning.

## 3. Methodology

In this section, we introduce **O**ne-**F**or-**A**ll **T**abular **A**nomaly **D**etection (OFA-TAD), a universal TAD approach capable of identifying anomalies across diverse tabular datasets without the need for dataset-specific training. The overall pipeline of OFA-TAD is demonstrated in Figure 3. Firstly, to align the tabular data from diverse domains, we introduce the **multi-view distance encoding module** (Sec. 3.1), which encodes heterogeneous features into multi-view neighbor distance profiles, where each view corresponds to a metric space induced by a specific feature transformation. Next, to adaptively select the optimal view for each sample, we design a **mixture-of-experts (MoE) scoring network** (Sec. 3.2), where each expert specializes in a specific metric space and a gating module dynamically weights their contributions. Finally, to address the absence of true anomaly labels during training, we introduce a **multi-strategy anomaly synthesis mechanism** (Sec. 3.3) that creates diverse pseudo-anomalies for supervised optimization.

### 3.1. Multi-View Distance Encoding

Tabular datasets from different domains vary significantly in feature dimensionality and semantics, making it non-trivial to directly feed raw features into a shared neural network. Moreover, the diversity of feature semantics across domains also hinders us from learning transferable semantic-level patterns for anomaly detection. To address these challenges, instead of attempting to align raw feature semantics, we leverage a domain-agnostic cue for TAD, i.e., anomalies are typically characterized by being less consistent with their local neighborhood. Based on this intuition, we represent each sample using its **neighbor distance patterns**, the distance distribution between a sample and its neighbors, which provides a unified input format across datasets.

**Neighbor Distance Extraction.** To convert variable-dimensional features into a fixed-length representation, we compute the distances to the top-$K$ nearest neighbors in the training set. For a sample $\mathbf{x}$, we retrieve its $K$ nearest neighbors from $\mathbf{X}_{train}$ and compute the Euclidean distances between $\mathbf{x}$ and its neighbors:

$$\mathbf{d} = [d_1, d_2, \ldots, d_K]^\top, \tag{1}$$

where $d_k$ denotes the distance to the $k$-th nearest neighbor. This yields a unified distance sequence regardless of the original feature dimensionality.

**Multi-View Transformation.** Although neighbor distance provides a unified token, a single distance metric is insufficient for capturing diverse anomaly patterns. In practice, different datasets usually exhibit specific preferences for particular metric spaces induced by feature transformations (as shown in Fig. 2b). This variability can be attributed to the heterogeneous structure and irregular feature distributions of

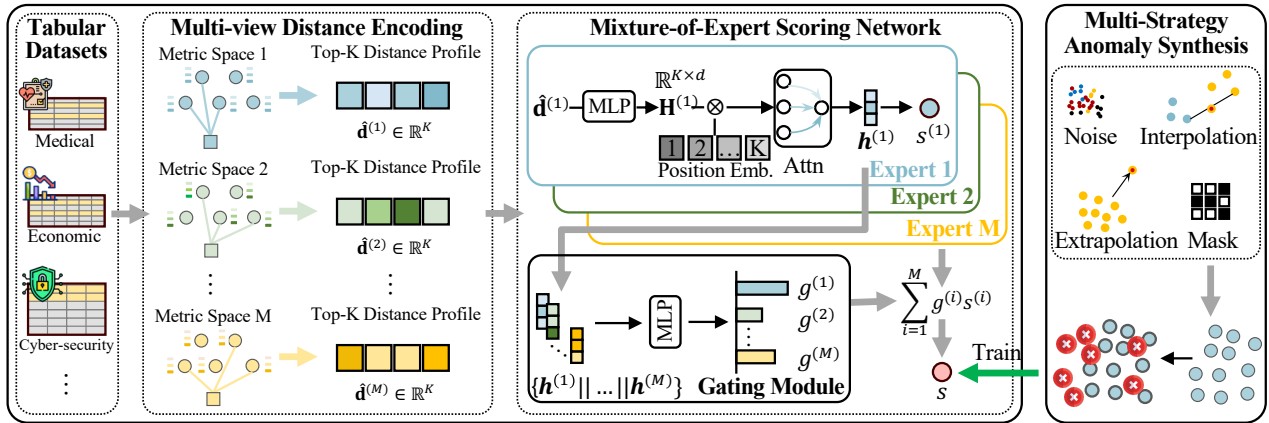

*Figure 3.* The pipeline of OFA-TAD. First, the *multi-view distance encoding module* extracts normalized neighbor-distance profiles from multiple transformation-induced metric spaces to obtain transferable representations. Then, the *Mixture-of-Experts (MoE) scoring network* leverages view-specific experts and an entropy-regularized gating mechanism to adaptively fuse distance-based anomaly evidence. A *multi-strategy anomaly synthesis module* is employed to generate diverse pseudo-anomalies for model training.

tabular data, which often challenge standard deep learning models and require tailored representations (Shwartz-Ziv & Armon, 2022; Grinsztajn et al., 2022; Beyazit et al., 2023). For example, some anomalies are detectable only in the standardized space, while others emerge in the rank-based space. To address this, we construct $M$ distinct *metric spaces* (i.e., the data views in OFA-TAD):

$$\mathbf{x}^{(m)} = \mathcal{T}_m(\mathbf{x}), \quad m \in \{1, \dots, M\}, \quad (2)$$

where each transformation $\mathcal{T}_m$ is fitted on the training set $\mathbf{X}_{train}$. Applying the above neighbor distance extraction in each view yields a view-specific distance sequence $\mathbf{d}^{(m)} = [d_1^{(m)}, d_2^{(m)}, \dots, d_K^{(m)}]^\top$, and thus multi-view distance sequences $\{\mathbf{d}^{(m)}\}_{m=1}^M$. By applying different transformations, the relative positions of samples in the feature space are reshaped, inducing different nearest neighbor structures and revealing complementary anomaly patterns.

**Distance Normalization.** Although the multi-view transformation provides a solution for capturing diverse distance-based anomaly patterns, the scales of raw distances can vary wildly across datasets (e.g., ranging from $10^{-2}$ to $10^5$), causing optimization instability and hindering cross-domain transferability. To eliminate such magnitude discrepancies, we apply a quantile normalization to map absolute distances into relative probabilities:

$$\hat{d}_k^{(m)} = \text{QuantileTransform}(d_k^{(m)}), \quad (3)$$

The quantile transformer is fitted on distances collected from the training set, mapping them into a uniform distribution $U[0, 1]$. Therefore, the distance-centric representation for each sample is the $M$ normalized distance sequences $\{\hat{\mathbf{d}}^{(m)}\}_{m=1}^M$, which serve as cross-domain comparable inputs for the downstream anomaly scoring network.

### 3.2. Mixture-of-Experts Scoring Network

The multi-view transformation generates a set of candidate distance patterns, yet they are *not equally reliable*: a transformation that improves separability on one dataset can be misleading on another (Fig. 2b). A naive fusion, such as concatenation or uniform averaging, ignores this view-specific reliability and may dilute the strongest signal with noise from suboptimal views, resulting in unstable cross-domain decisions. To address this issue, we employ a Mixture-of-Experts (MoE) scoring network for sample-adaptive fusion. Concretely, each view is processed by a dedicated expert that extracts view-specific evidence and outputs a view-wise score, while a gating module predicts sample-specific mixture weights. The mixture emphasizes informative views and suppresses distracting ones, producing a single anomaly score suitable for heterogeneous datasets.

**Positional Embedding.** The neighbor-distance profile is an ordered sequence (from the closest to the farthest neighbor), where early ranks often capture local density changes that are crucial for anomaly detection. If the expert ignores this ordering, it may treat near and far neighbors as interchangeable and blur the distinction between informative local evidence and less informative distant context. To address this, we first project each distance value into a $D$-dimensional token and then inject learnable positional embeddings to encode the neighbor rank:

$$\mathbf{H}^{(m)} = \text{LayerNorm}(\text{MLP}_{\text{enc}}^{(m)}(\hat{\mathbf{d}}^{(m)})) + \mathbf{P}_{pos}, \quad (4)$$

where $\mathbf{H}^{(m)} \in \mathbb{R}^{K \times D}$ and $\mathbf{P}_{pos}$ encodes the rank information of each neighbor. This produces rank-aware neighbor representations, allowing the expert to distinguish deviations among the nearest neighbors from those occurring only in the distance tail.

**Attention Pooling.** Although positional encoding makes

neighbor ranks identifiable, the contribution of each rank to anomaly judgment remains inherently sample-dependent. Consequently, a fixed aggregation rule such as mean pooling can blur discriminative distance evidence by indiscriminately mixing informative and uninformative neighbors. Instead of using a fixed rule, we employ attention pooling to learn a content-dependent aggregation over neighbor ranks:

$$\alpha_k^{(m)} = \text{Softmax}({\mathbf{w}_{\text{att}}^{(m)}}^\top \sigma(\mathbf{W}_{\text{att}}^{(m)} \mathbf{H}_k^{(m)})), \quad (5)$$

$$\mathbf{h}^{(m)} = \sum_{k=1}^{K} \alpha_k^{(m)} \mathbf{H}_k^{(m)}, \quad (6)$$

where $\mathbf{w}_{\text{att}}^{(m)}$ and $\mathbf{W}_{\text{att}}^{(m)}$ are learnable parameters, and the attention weights $\{\alpha_k\}_{k=1}^K$ quantify how informative each neighbor rank is for the current sample under view $m$. This allows the model to focus on a few critical neighbors and suppress irrelevant ones. After this step, we obtain a compact view-level embedding $\mathbf{h}^{(m)}$ for each view. The attention pooling mechanism ensures the expert representation is more robust to noisy or uninformative neighbors, improving the signal-to-noise ratio of the distance evidence and facilitating stable cross-domain scoring.

*Discussion:* While both the positional embedding and attention pooling focus on modeling rank-aware neighborhood information, they are complementary to each other. Position embedding provides a structural prior that the $k$-th neighbor rank has a consistent meaning across samples (for instance, early ranks reflect local density), which stabilizes the learning process. Attention pooling provides sample adaptivity by selecting which ranks are most indicative for the current sample under a given view. Together, they produce $\mathbf{h}^{(m)}$ that is rank-aware and sample-specific, yielding a more robust representation for downstream anomaly scoring.

**Expert Scoring.** The resulting view-level embedding $\mathbf{h}^{(m)}$ serves as a compact summary of the sample's distance profile under view $m$. Based on $\mathbf{h}^{(m)}$, each expert outputs an anomaly score through an MLP-based scoring network:

$$s^{(m)} = \text{MLP}_{\text{score}}^{(m)}(\mathbf{h}^{(m)}), \quad (7)$$

where $s^{(m)}$ measures how anomalous the sample appears at view $m$. The score $s^{(m)}$, as the final output of each view expert, reflects the distance-based anomaly evidence in the metric space induced by a specific transformation.

**Gating-based Mixture Prediction.** Although each view provides an anomaly score from a transformation-specific metric space, its reliability for judging abnormality can vary across datasets and even across individual samples. In OFA-TAD, we introduce a gating module that observes all expert embeddings and predicts view-specific weights to dynamically determine the reliability of each view:

$$\mathbf{g} = \text{Softmax}\Big(\text{MLP}_{\text{gate}}\big(\text{Concat}([\mathbf{h}^{(1)}, \ldots, \mathbf{h}^{(M)}])\big)\Big), \quad (8)$$

where the $m$-th element in $\mathbf{g} \in \mathbb{R}^M$, $g_m$, indicates the importance weight assigned to view $m$. By conditioning on embeddings instead of raw distances, the gating module can judge view reliability from high-level distance evidence, such as whether the profile appears "clean" or "separable", enabling finer-grained sample-wise view fusion.

With the predicted weights, we aggregate expert scores into the final anomaly score:

$$s = \sigma \left( \sum_{m=1}^{M} g_m \cdot s^{(m)} \right). \quad (9)$$

This mixture form implements data-specific view selection: for a sample from any unknown target domain, OFA-TAD can assign higher weights to informative views when computing the final score, which reduces the sensitivity to transformation-induced distance variations.

### 3.3. Training with Multi-Strategy Anomaly Synthesis

TAD is typically treated as a one-class classification problem due to the lack of true anomalies during training (Han et al., 2022). This makes optimization challenging: purely one-class objectives can be unstable and may fail to learn a sharp boundary in complex data manifolds, as observed in DeepSVDD (Ruff et al., 2018), where hypersphere collapse can occur. To provide explicit supervision while respecting the one-class constraint, we synthesize pseudo-anomalies and recast training as a binary classification problem. Our core idea is to expose OFA-TAD to diverse pseudo-abnormal patterns, thereby encouraging the model to learn a more discriminative decision function.

**Multi-Strategy Anomaly Synthesis.** Real-world anomalies are diverse, including global outliers, local outliers, and boundary cases, so a single synthesis rule provides limited coverage and can bias the learned boundary. We therefore design a mixed generator that integrates four complementary strategies to create a broader spectrum of pseudo-anomalies, including ❶ **Manifold Extrapolation**, to simulate anomalies beyond the data manifold boundary: $\mathbf{x}_{neg} = \mathbf{x}_b + \alpha(\mathbf{x}_b - \mathbf{x}_a)$, where $\alpha \geq 0$; ❷ **Inter-Cluster Interpolation**, to generate samples in low-density regions between clusters: $\mathbf{x}_{neg} = \beta\mathbf{x}_a + (1-\beta)\mathbf{x}_b$, where $\mathbf{x}_a$ and $\mathbf{x}_b$ belong to different clusters; ❸ **Noise Injection**, to add Gaussian or uniform noise to simulate measurement errors; and ❹ **Feature Masking**, to randomly masks features to simulate data corruption.

**Model Training.** Using the synthesized anomalies (with $y_i = 1$) and normal training samples (with $y_i = 0$), we

*Table 1.* AUROC comparison between baselines and OFA-TAD. The best results are highlighted in **bold**.

| Dataset | LOF | KNN | iForest | DSVDD | AE | MCM | LUNAR | DRL | DisentAD | OFA-TAD |
|---|---|---|---|---|---|---|---|---|---|---|
| *In-Domain Target Datasets* | | | | | | | | | | |
| abalone | 0.7284 | 0.7997 | 0.7371 | 0.6756 | 0.8014 | 0.7450 | 0.8084 | 0.8071 | 0.7789 | **0.8178** |
| arrhythmia | 0.8075 | 0.8014 | 0.7931 | 0.7566 | 0.7260 | 0.7848 | 0.7623 | 0.7467 | 0.7567 | **0.8095** |
| breastw | 0.9776 | 0.9764 | 0.9975 | 0.9913 | 0.9778 | **0.9977** | 0.9836 | 0.9949 | 0.9960 | 0.9791 |
| cardio | 0.9374 | 0.9062 | 0.9352 | **0.9571** | 0.9500 | 0.8710 | 0.9121 | 0.9431 | 0.9480 | 0.9322 |
| census | 0.5451 | 0.6751 | 0.6364 | 0.6916 | 0.6916 | 0.6696 | 0.6565 | 0.5914 | **0.8345** | 0.6955 |
| donors | 0.9845 | 0.9991 | 0.9029 | 0.7494 | 0.9384 | 0.9965 | 0.9979 | 0.9002 | 0.9073 | **0.9997** |
| fault | 0.4742 | 0.5873 | 0.5665 | 0.5163 | 0.5818 | **0.6165** | 0.5340 | 0.5910 | 0.6144 | 0.5762 |
| Hepatitis | 0.6131 | 0.4910 | 0.7226 | **0.7837** | 0.5656 | 0.6108 | 0.5534 | 0.6855 | 0.6371 | 0.7353 |
| lympho | 0.9577 | 0.9343 | 0.9972 | **0.9981** | 0.9413 | 0.9915 | 0.9828 | 0.9920 | 0.8338 | 0.9911 |
| mammography | 0.8599 | 0.8724 | 0.8835 | 0.8615 | 0.8935 | **0.9073** | 0.8719 | 0.8788 | 0.8880 | 0.9000 |
| mnist | 0.9511 | 0.9348 | 0.8657 | 0.8434 | 0.9215 | 0.9640 | 0.8796 | **0.9645** | 0.5835 | 0.9433 |
| musk | **1.0000** | **1.0000** | 0.9748 | 0.9825 | **1.0000** | 0.9966 | 0.9975 | 0.9999 | 0.9690 | **1.0000** |
| optdigits | **0.9936** | 0.9680 | 0.8020 | 0.5636 | 0.8535 | 0.9891 | 0.9329 | 0.8225 | 0.9926 | 0.9930 |
| Parkinson | 0.6967 | 0.4615 | **0.7718** | 0.6840 | 0.6936 | 0.3379 | 0.4746 | 0.6603 | 0.7101 | 0.7164 |
| pendigits | 0.9871 | 0.9883 | 0.9642 | 0.7754 | 0.9554 | 0.9842 | 0.9897 | 0.9391 | 0.9932 | **0.9990** |
| pima | 0.6609 | 0.6807 | 0.7331 | 0.6898 | 0.7282 | 0.7131 | **0.7349** | 0.7313 | 0.7161 | 0.6959 |
| satimage-2 | 0.9938 | 0.9971 | 0.9910 | 0.9730 | 0.9985 | **0.9986** | 0.9517 | 0.9857 | 0.8515 | 0.9964 |
| shuttle | 0.9972 | 0.9964 | 0.9964 | 0.9934 | 0.9976 | 0.9986 | 0.9759 | 0.9983 | 0.9993 | **0.9998** |
| thyroid | 0.9271 | 0.9868 | **0.9886** | 0.9851 | 0.9696 | 0.9418 | 0.9666 | 0.9830 | 0.9880 | 0.9809 |
| wbc | 0.9715 | 0.9707 | 0.9626 | 0.9448 | 0.9582 | 0.9708 | 0.9559 | **0.9821** | 0.9777 | 0.9516 |
| WDBC | 0.9989 | 0.9978 | 0.9980 | 0.9868 | 0.9961 | 0.9918 | 0.9870 | 0.9990 | 0.9977 | **0.9996** |
| WPBC | 0.5062 | 0.4801 | 0.4973 | 0.4776 | 0.4720 | 0.5093 | 0.5039 | 0.5100 | **0.5567** | 0.4830 |
| yeast | 0.4571 | 0.4450 | 0.4182 | 0.4546 | 0.4640 | 0.4454 | **0.5668** | 0.4786 | 0.4988 | 0.4656 |
| *Out-of-Domain Target Datasets* | | | | | | | | | | |
| amazon | 0.5392 | 0.5387 | 0.5080 | 0.5005 | 0.5282 | 0.5201 | 0.5177 | 0.5070 | 0.5465 | **0.5469** |
| backdoor | 0.9381 | 0.9358 | 0.7481 | 0.6724 | 0.9250 | 0.9678 | 0.9295 | **0.9796** | 0.6970 | 0.9592 |
| campaign | 0.6437 | 0.7407 | 0.7374 | 0.7091 | 0.7753 | **0.8354** | 0.6760 | 0.7308 | 0.7846 | 0.7564 |
| cover | 0.9121 | 0.8796 | 0.7475 | 0.8335 | 0.8630 | 0.9201 | 0.9472 | **0.9872** | 0.9850 | 0.9627 |
| fraud | 0.3569 | 0.9317 | **0.9402** | 0.9398 | 0.9365 | 0.9270 | 0.8457 | 0.8311 | 0.9298 | 0.8785 |
| glass | 0.5771 | 0.5663 | 0.5676 | 0.5441 | 0.5717 | 0.5804 | 0.5946 | 0.5853 | **0.8996** | 0.6690 |
| ionosphere | 0.8344 | 0.9490 | 0.8419 | 0.8919 | 0.9605 | 0.9676 | 0.9578 | 0.9671 | **0.9690** | 0.9639 |
| SpamBase | 0.7323 | 0.7700 | 0.8474 | 0.7796 | 0.8180 | 0.7461 | 0.7973 | 0.8380 | 0.6025 | **0.8599** |
| speech | 0.3788 | 0.3666 | 0.3997 | 0.4328 | 0.3667 | 0.4470 | **0.5628** | 0.5595 | 0.5493 | 0.4891 |
| vowels | 0.8487 | 0.8221 | 0.5925 | 0.5015 | 0.8102 | 0.8539 | 0.8346 | 0.8500 | **0.9283** | 0.8151 |
| Wilt | 0.6881 | 0.7545 | 0.4816 | 0.3892 | 0.4419 | 0.7485 | 0.7827 | 0.7790 | 0.7543 | **0.8102** |
| Average | 0.7787 | 0.8001 | 0.7808 | 0.7509 | 0.7963 | 0.8102 | 0.8066 | 0.8176 | 0.8140 | **0.8345** |

train OFA-TAD in an end-to-end manner to predict anomaly scores with a Mean Squared Error (MSE) loss:

$$\mathcal{L} = \frac{1}{n_{train}} \sum_{i=1}^{n_{train}} (s_i - y_i)^2, \qquad (10)$$

where $n_{train}$ represents the number of training samples, including all samples in $\mathcal{P}_{train}$ and their corresponding synthetic pseudo-anomalies. Trained by the normal samples from various training datasets and diverse pseudo-anomalies synthesized by multiple negative sampling strategies, OFA-TAD can learn a transferable decision boundary and hence generalize to unseen domains. The training and inference algorithmic description of OFA-TAD is provided in Appendix B, with complexity analysis given in Appendix C.

## 4. Experiments

### 4.1. Experiments Setup

**Datasets.** OFA-TAD are trained on 7 datasets and tested on 34 datasets, sourced from ADBench (Han et al., 2022), with detailed statistics in Appendix D. Each dataset is assigned a semantic category. We adopt a category-based split to reflect the One-for-All (OFA) setting. For any category with at least three datasets, we select one dataset as a *source* dataset and treat the remaining datasets within the same category as *in-domain* test datasets. For the Healthcare category, we select two datasets as source datasets due to its large coverage. Categories with fewer than three datasets are treated as out-of-domain target datasets.

**Baselines.** We compare OFA-TAD with 9 representative TAD methods, including *classic methods*, i.e., Isola-

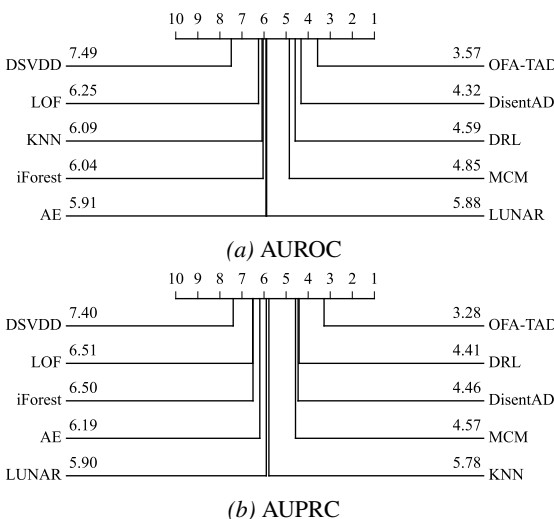

*(a) AUROC*

*(b) AUPRC*

*Figure 4.* Average rank of different methods across 34 datasets.

tion Forest (IForest) (Liu et al., 2008), Local Outlier Factor (LOF) (Breunig et al., 2000), and $k$NN-based detection (KNN) (Angiulli & Pizzuti, 2002), as well as *deep learning-based approaches*, i.e., AutoEncoder (AE) (Chen et al., 2018), DeepSVDD (DSVDD) (Liznerski et al., 2021), LUNAR (Goodge et al., 2022), MCM (Yin et al., 2024), DRL (Ye et al., 2025a), and DisentAD (Ye et al., 2025b).

**Evaluation Metrics and Implementation.** We employ AUROC and AUPRC as primary metrics. We also report F1 score under a consistent thresholding protocol. To summarize performance across datasets with different scales, we report the average rank across target datasets. We train OFA-TAD once on the source datasets for 15 epochs using the Adam optimizer with a learning rate of $5 \times 10^{-4}$ and weight decay of $2 \times 10^{-5}$, without any learning rate schedulers. During training and inference, we extract $K = 80$ nearest neighbors for each view. In the MoE network, the embedding dimension is set to 128, and each expert is parameterized by a 2-layer MLP with a hidden dimension of 64. Importantly, we keep these hyperparameters fixed across all datasets to avoid per-dataset tuning. During inference, we exclusively use the training split of each target dataset as domain context for neighborhood retrieval and distance normalization. For all baselines, we evaluate them under the OFO paradigm, i.e., training a separate model for each target dataset. Additional details on baseline evaluation under the OFA protocol are provided in Appendix E.

### 4.2. Performance Comparison

Table 1 summarizes AUROC of OFA-TAD and all baselines on target datasets, and AUPRC/F1 results are deferred to Appendix F.1. We have observations: ❶ The winners are relatively distributed: different methods achieve the best performance on different datasets, reflecting the strong

heterogeneity of tabular domains and suggesting that no single inductive bias dominates all target datasets. ❷ OFA-TAD achieves the best overall performance across all target datasets, indicating consistently strong performance beyond win counts. ❸ Even on the out-of-domain block, OFA-TAD remains competitive under domain shift. Notably, unlike baselines that are trained and evaluated within each dataset, OFA-TAD never retrains or performs dataset-specific tuning on target datasets; it only uses the training split of each target dataset as context during inference. The consistent advantage under this strict setting highlights the effectiveness of context-based neighborhood modeling for cross-domain TAD.

Figure 4 further visualizes the average ranks w.r.t. AUROC and AUPRC (F1 results are in Appendix F.2). OFA-TAD consistently stays at the top across all three metrics, indicating that its advantage is not tied to a particular operating point but remains stable under both ranking-based (AUROC/AUPRC) and threshold-dependent (F1) evaluation. In contrast, several baselines exhibit noticeable rank fluctuations across metrics, suggesting metric-sensitive behavior and less reliable cross-dataset generalization.

### 4.3. Ablation Studies

To validate the effectiveness of each design in OFA-TAD, we construct several variants: ❶ **w/o Gating** that uses uniform fusion over views, ❷ **w/o MoE** that removes experts and use non-parametric pooling, ❸ **w/o Attention** that replaces attention pooling with mean pooling, and ❹ **w/o Position** that removes positional embeddings. The performance comparison w.r.t. average AUROC/AUPRC/F1 is summarized in the upper part of Table 2 (full per-dataset results are provided in Appendix H). Overall, all components contribute to performance, since removing any module reduces the performance. In particular, removing attention pooling causes the largest drop, suggesting that explicitly weighting neighbor evidence is crucial for highlighting informative neighbors when anomaly signals are sparse.

**Ablation on Synthesis Strategy.** Negative sample generation is a key ingredient in training OFA-TAD. To verify the effectiveness of each pseudo-anomaly synthesis strategy, we remove one strategy at a time and measure performance change. As shown in the bottom part of Table 2, using all strategies yields the best AUROC. Removing any single strategy degrades performance, indicating these strategies provide complementary supervision signals. In particular, removing noise injection or manifold extrapolation leads to the largest drop, while removing inter-cluster interpolation or feature masking results in smaller degradation.

*Table 2.* Ablation results w.r.t. average AUROC, AUPRC, and F1.

| Variant | AUROC | AUPRC | F1 |
|---|---|---|---|
| OFA-TAD | **0.8345** | **0.6629** | **0.6352** |
| w/o Gating | 0.8218 | 0.6498 | 0.6211 |
| w/o MoE | 0.8204 | 0.6448 | 0.6177 |
| w/o Attention | 0.8187 | 0.6383 | 0.6029 |
| w/o Position | 0.8281 | 0.6404 | 0.6124 |
| w/o Noise Inject | 0.8203 | 0.6011 | 0.5788 |
| w/o Extrapolation | 0.8190 | 0.6061 | 0.5781 |
| w/o Interpolation | 0.8317 | 0.6614 | 0.6349 |
| mw/o Masking | 0.8304 | 0.6484 | 0.6154 |

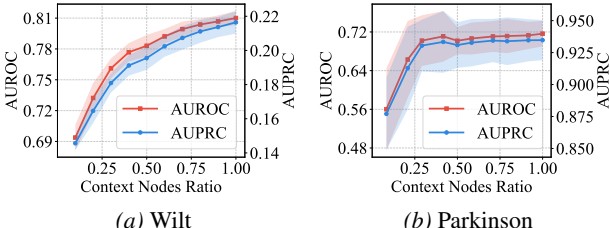

*(a)* Wilt      *(b)* Parkinson

*Figure 5.* Performance with varying context nodes.

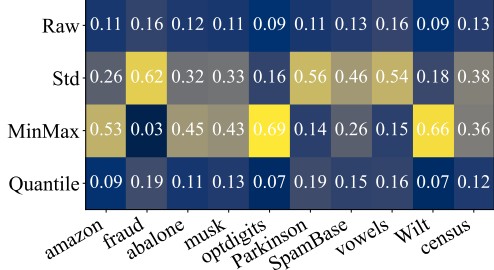

*Figure 6.* Gating weight heatmaps across datasets.

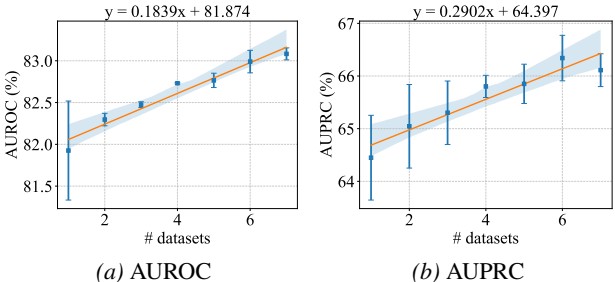

*(a)* AUROC      *(b)* AUPRC

*Figure 7.* model scaling laws on data size.

## 4.4. Robustness to Varying Context Size

We vary the number of context samples in target datasets (from proportion 0.1 to 1.0) to evaluate robustness under limited context size. As shown in Figure 5, on Wilt, both AUROC and AUPRC improve steadily as the context ratio increases, with the largest gain appearing when moving from very limited context to a moderate amount and then gradually saturating. On Parkinson, performance rises sharply with only a small amount of context (around 0.3) and then plateaus, suggesting that a modest context set is sufficient for stable inference while additional context yields diminishing returns. Moreover, the uncertainty bands shrink as the context ratio increases, indicating stable predictions when more target-domain neighbors are available. These results demonstrate that OFA-TAD can perform reliable on-the-fly inference even when only a small subset of target-domain normal samples is available as context. Additional context-size results on more datasets are provided in Appendix I.

## 4.5. Visualization

To interpret model behavior, we visualize the mean gating weights over the four metric views, i.e., the raw feature space (Raw), the standardized space (Std), the min–max normalized space (MinMax), and the quantile-transformed space (Quantile). From Figure 6 we can see the learned weights exhibit clear dataset-dependent preferences, where the gate often concentrates on one dominant view: for example, Std receives high weights on datasets such as *fraud* and *Parkinson*, whereas MinMax is favored on datasets such as

*amazon*, *optdigits*, and *Wilt*. In contrast, Raw and Quantile are consistently assigned small weights across most datasets, suggesting that they are less informative for distance-based separability under our unified protocol. Overall, this visualization supports our motivation that the optimal distance view varies across domains and that the gating module enables OFA-TAD to adaptively select and fuse informative metric spaces at inference time. Additional gating visualizations on more target datasets are provided in Appendix J.

## 4.6. Scaling with #Source Datasets

We ablate the number of source datasets used for pre-training and evaluate the resulting OFA transfer performance on all target datasets under the same protocol. To avoid enumerating a combinatorial space of source subsets, we progressively include the first $k$ source datasets in our split and report the mean performance over target datasets. Figure 7 shows a clear scaling trend: using more pre-training datasets improves transfer performance, with a near-linear upward trend for both AUROC and AUPRC and only minor fluctuations at larger $k$. We further fit an ordinary least squares line to quantify this relationship. On average, adding one more source dataset leads to a 0.18 AUROC (%) gain and a 0.29 AUPRC (%) gain. The gains are larger when moving from very few sources to a moderate number of sources and then gradually saturate, suggesting diminishing returns once the pre-training pool becomes sufficiently diverse. Also, the uncertainty bands remain small across $k$, indicating that the scaling trend is stable under multiple runs. A broader analysis on random source combinations is provided in Appendix G, which further confirms the strong robustness of

OFA-TAD to arbitrary source dataset selection.

## 5. Related Work

**Tabular Data Anomaly Detection.** Tabular anomaly detection (TAD) is traditionally tackled under a one-for-one (OFO) paradigm, requiring a dedicated model to be trained and tuned for each specific dataset. Classical methods rely on fixed inductive biases like density and distance in the original feature space (Liu et al., 2008; Breunig et al., 2000). Deep learning approaches advance TAD by utilizing reconstruction and self-supervised objectives to capture intrinsic data regularities (Chen et al., 2018; Qiu et al., 2021; Ye et al., 2025a). However, their reliance on dataset-specific optimization limits their transferability across heterogeneous domains. In contrast, our OFA-TAD targets the one-for-all (OFA) setting, discovering transferable anomaly patterns based on neighbor-distance representations without target-domain retraining.

**One-for-All Anomaly Detection.** An emerging direction across various domains is OFA modeling, where a single pretrained foundation model generalizes to unseen datasets. Vision and graph models adapt to new tasks via prompt-based interfaces and in-context learning (Chen et al., 2023; Zhu & Pang, 2024; Liu et al., 2024), while time-series models pretrain on massive corpora and use adaptive bottlenecks to handle cross-domain divergence (González et al., 2025; Shentu et al., 2025). A shared principle among these generalist methods is that effective transfer relies on robust representations coupled with test-time adaptation. Our work brings this principle to the tabular domain by representing samples through multi-view distance patterns as domain-agnostic tokens, and employing sample-adaptive fusion for on-the-fly detection. A more detailed discussion of related work can be found in Appendix A.

## 6. Conclusions

In this paper, we take a first step toward one-for-all (OFA) tabular anomaly detection by proposing OFA-TAD, a generalist detector that transfers to unseen tabular datasets without dataset-specific retraining or fine-tuning. OFA-TAD leverages normalized multi-view neighbor-distance as transferable patterns and a mixture-of-experts model for sample-adaptive scoring, enabling on-the-fly inference using only normal context samples from the target dataset. Extensive experiments on 34 real-world datasets from 14 domains demonstrate the effectiveness and generalizability of OFA-TAD under the strict OFA protocol, and we further observe a consistent scaling trend where using more source datasets for pre-training improves transfer performance on average. Building on these findings, **future work** includes scaling to larger source collections and improving view generation,

routing, and context construction for more efficient and robust deployment.

## Acknowledgements

The work of S. Pan was partially supported by the Australian Research Council (ARC) under Grant Nos. DP240101547 and FT210100097. The work of Y. Liu was partially supported by the ARC under Grant No. DE260101172.

## Impact Statement

This paper advances tabular anomaly detection by proposing a one-for-all framework that can be trained once and transferred to new tabular datasets without target-domain retraining or hyperparameter tuning. Such a capability may reduce the computational and operational burden of deploying anomaly detectors across heterogeneous real-world domains (e.g., health monitoring, fraud detection, and system reliability), potentially enabling faster iteration and broader access to anomaly detection tools.

At the same time, anomaly detection systems can have negative consequences if deployed without care. False positives may trigger unnecessary interventions, audits, or service disruptions, while false negatives may delay the discovery of critical events; these risks can be amplified under domain shift. In addition, the method uses target-domain training data as inference-time context, which requires responsible data governance to avoid privacy leakage or inappropriate secondary use of sensitive information. Overall, we do not identify any significant risks uniquely introduced by OFA-TAD beyond the standard considerations common to anomaly detection systems, and potential risks largely depend on the deployment context and downstream decisions. We recommend deploying such models with domain-appropriate safeguards, including human oversight for high-stakes decisions, calibration/threshold selection aligned with application costs, and monitoring for dataset-dependent failure modes and potential bias across subpopulations.

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

# Appendix

# A. Detailed Related Work

## A.1. Tabular Data Anomaly Detection

Tabular anomaly detection (TAD) is often studied under the one-class setting, where only normal samples are available during training and anomalies are identified as deviations at test time (Han et al., 2022). Classical methods build on fixed inductive biases in the original feature space, including isolation-based approaches such as Isolation Forest (Liu et al., 2008), local-density and distance-based methods such as LOF (Breunig et al., 2000) and $k$NN scoring (Angiulli & Pizzuti, 2002), and reconstruction-based techniques that project data to a low-dimensional subspace and flag samples with large reconstruction error (Chen et al., 2018). These methods are simple and training-efficient, but their reliance on handcrafted geometry and preprocessing makes it difficult to maintain stable performance across heterogeneous domains.

Recent years have witnessed substantial progress in deep learning-based TAD (Thimonier et al., 2024). A common line adopts reconstruction objectives to learn normal patterns and uses reconstruction error as anomaly scores (Chen et al., 2018). Beyond reconstruction, representation learning and self-supervised objectives have been explored to capture intrinsic regularities of normal data, including contrastive learning and invariance-inducing objectives (e.g., NeuTraLAD) (Qiu et al., 2021), masked modeling for tabular data (e.g., MCM) (Yin et al., 2024), and disentanglement-inspired learning (e.g., DRL and DisentAD) (Ye et al., 2025a;b).

Despite strong in-domain performance, most existing methods are still developed and tuned under the conventional one-for-one (OFO) paradigm, requiring dataset-specific training and hyperparameter search for each new dataset. In contrast, our work targets the one-for-all (OFA) setting and discovers transferable, domain-agnostic anomaly patterns based on neighbor-distance representations. By combining multi-view distance encoding with sample-adaptive fusion, our model can leverage normal samples from the target dataset as inference-time context to perform on-the-fly anomaly detection without target-domain retraining.

## A.2. One-for-All Anomaly Detection

An emerging direction in anomaly detection is OFA modeling, where a single pretrained model is expected to generalize across datasets and even across application domains (Liu et al., 2026; Pan et al., 2025; 2026a;b). Rather than retraining for each target task, these approaches typically rely on foundation backbones and shift adaptation to inference time through prompts, context examples, or lightweight conditioning mechanisms (Li et al., 2026a; Miao et al., 2025; Li et al., 2024; 2026b).

In computer vision, a representative line of work leverages vision–language models (VLMs) as universal feature extractors and adapts them via prompt-based interfaces (Zhou et al., 2024). WinCLIP reformulates industrial anomaly detection by aligning visual features at the window, patch, and image-level with compositional text prompts in CLIP, enabling zero-shot and few-normal-shot anomaly classification and localization without task-specific training (Chen et al., 2023). Beyond VLMs, generative foundation models have also been explored for zero-shot anomaly detection: diffusion-based methods repurpose denoising trajectories as perceptual templates and derive anomaly scores from reconstruction or score-matching errors (Abdi et al., 2025). Another thread aims to reduce reliance on handcrafted prompts by treating target-domain context itself as part of the input. InCTRL formulates generalist anomaly detection as an in-context residual learning problem, where a query sample is evaluated by comparing its residual patterns against a small set of normal examples provided at test time (Zhu & Pang, 2024).

For video anomaly detection, MDVAD investigates multi-domain learning and reveals that conflicting definitions of abnormality across datasets can hinder transfer, motivating explicit modeling of cross-domain ambiguity during training (Cho et al., 2024; Tan et al., 2025). Motivated by broader anomaly coverage, AnomalyMoE proposes a language-free generalist framework based on a mixture-of-experts architecture, decomposing anomalies into local structural, component-level semantic, and global logical hierarchies (Gu et al., 2026). By assigning dedicated experts to each semantic level and encouraging expert diversity, it aims to unify heterogeneous anomaly types within a single model.

Generalist anomaly detection has also been studied beyond vision. In graph domains, ARC adopts an in-context learning paradigm that extracts dataset-specific regularities from a few normal nodes at inference time, combining feature alignment, residual encoding, and cross-attentive scoring for on-the-fly adaptation (Liu et al., 2024; 2026; Zhao et al., 2026). Complementarily, AnomalyGFM pretrains a graph foundation model by aligning learnable normal and abnormal prototypes with

---

**Algorithm 1** The training algorithm of OFA-TAD (one-time pre-training on source datasets)

---

**Require:** Source training pool $\mathbf{X}_{src}$; views $\{\mathcal{T}_m\}_{m=1}^M$; neighbors $K$; epochs $T$; batch size $B$
**Ensure:** Trained MoE parameters $\Theta$
 1: Fit each $\mathcal{T}_m$ on $\mathbf{X}_{src}$
 2: Collect view-wise neighbor distances on $\mathbf{X}_{src}$ and fit quantile normalization (Eq. 3)
 3: Initialize MoE parameters $\Theta$
 4: **for** $t = 1$ to $T$ **do**
 5:     Sample a mini-batch of normal samples $\mathcal{B} \subset \mathbf{X}_{src}$ with $|\mathcal{B}| = B$
 6:     Generate pseudo anomalies $\mathcal{B}_{neg}$ from $\mathcal{B}$ using the multi-strategy generator
 7:     Form training batch $\tilde{\mathcal{B}} = \mathcal{B} \cup \mathcal{B}_{neg}$ with labels $y \in \{0, 1\}$
 8:     **for all** $\mathbf{x} \in \tilde{\mathcal{B}}$ **do**
 9:         **for** $m = 1$ to $M$ **do**
10:             Compute $\mathbf{x}^{(m)} = \mathcal{T}_m(\mathbf{x})$
11:             Retrieve Top-$K$ neighbors in view $m$ and compute $\mathbf{d}^{(m)}(\mathbf{x})$
12:             Normalize distances to obtain $\hat{\mathbf{d}}^{(m)}(\mathbf{x})$ (Eq. 3)
13:         **end for**
14:         Compute expert embeddings/scores and gating weights (Eq. 8); obtain $s(\mathbf{x})$ (Eq. 9)
15:     **end for**
16:     Compute prediction loss and update $\Theta$ by gradient descent (Sec. 3.3)
17: **end for**

---

**Algorithm 2** Inference of OFA-TAD under the OFA protocol on a target dataset

---

**Require:** Target training split $\mathbf{X}_{train}^{tgt}$ (context); target test split $\mathbf{X}_{test}^{tgt}$; trained parameters $\Theta$; views $\{\mathcal{T}_m\}_{m=1}^M$; neighbors $K$
**Ensure:** Anomaly scores $\{s(\mathbf{x})\}_{\mathbf{x} \in \mathbf{X}_{test}^{tgt}}$
 1: Fit each $\mathcal{T}_m$ on $\mathbf{X}_{train}^{tgt}$
 2: Build a neighbor index over $\mathbf{X}_{train}^{tgt}$ in each view (exact $k$NN or ANN)
 3: **for all** $\mathbf{x} \in \mathbf{X}_{test}^{tgt}$ **do**
 4:     **for** $m = 1$ to $M$ **do**
 5:         Compute $\mathbf{x}^{(m)} = \mathcal{T}_m(\mathbf{x})$
 6:         Retrieve Top-$K$ neighbors in view $m$ and compute $\mathbf{d}^{(m)}(\mathbf{x})$
 7:         Normalize distances to obtain $\hat{\mathbf{d}}^{(m)}(\mathbf{x})$ (Eq. 3)
 8:     **end for**
 9:     Forward $\{\hat{\mathbf{d}}^{(m)}(\mathbf{x})\}_{m=1}^M$ through the MoE to obtain $s(\mathbf{x})$
10: **end for**

---

neighborhood-based residual representations, enabling zero-shot inference and few-shot prompt tuning across diverse graph datasets (Qiao et al., 2025; Chen et al., 2025).

Extending the generalization challenge to time series data, recent work similarly pushes toward pretrained models that can be reused across domains. Foundation Auto-Encoders (FAE) pretrain a variational AE (VAE)-based generative model on massive time-series corpora and combine VAEs with dilated convolutional architectures to capture complex temporal patterns, aiming to support out-of-the-box modeling and zero-shot detection on previously unseen datasets (González et al., 2025). In a complementary direction, DADA constructs a general time-series anomaly detector by pretraining on multi-domain data and explicitly addressing cross-domain divergence through adaptive bottlenecks and dual adversarial decoders, which helps flexibly control information flow while separating multiple normal and abnormal patterns in downstream scenarios (Shentu et al., 2025).

Overall, these studies highlight a shared trend: effective transfer stems less from per-dataset optimization and more from robust representations coupled with test-time adaptation interfaces. Our work follows this principle in the tabular domain by representing samples through multi-view neighbor-distance patterns as domain-agnostic input tokens, and by performing sample-adaptive fusion to support on-the-fly anomaly detection.

*Table 3.* The statistics of in-domain datasets.

| Dataset | Category | Train | Test | # Samples | # Features | # Anomaly | % Anomaly |
|---|---|:---:|:---:|---:|---:|---:|---:|
| **In-domain datasets** | | | | | | | |
| satellite | Astronautics | ✓ | - | 6435 | 36 | 2036 | 31.64 |
| satimage-2 | Astronautics | - | ✓ | 5803 | 36 | 71 | 1.22 |
| shuttle | Astronautics | - | ✓ | 49097 | 9 | 3511 | 7.15 |
| vertebral | Biology | ✓ | - | 240 | 6 | 30 | 12.50 |
| yeast | Biology | - | ✓ | 1484 | 8 | 507 | 34.16 |
| abalone | Biology | - | ✓ | 4177 | 7 | 2081 | 49.82 |
| annthyroid | Healthcare | ✓ | - | 7200 | 6 | 534 | 7.42 |
| breastw | Healthcare | - | ✓ | 683 | 9 | 239 | 34.99 |
| cardio | Healthcare | - | ✓ | 1831 | 21 | 176 | 9.61 |
| Cardiotocography | Healthcare | ✓ | - | 2114 | 21 | 466 | 22.04 |
| Hepatitis | Healthcare | - | ✓ | 80 | 19 | 13 | 16.25 |
| lympho | Healthcare | - | ✓ | 148 | 18 | 6 | 4.05 |
| mammography | Healthcare | - | ✓ | 11183 | 6 | 260 | 2.32 |
| Pima | Healthcare | - | ✓ | 768 | 8 | 268 | 34.90 |
| thyroid | Healthcare | - | ✓ | 3772 | 6 | 93 | 2.47 |
| WDBC | Healthcare | - | ✓ | 367 | 30 | 10 | 2.72 |
| WPBC | Healthcare | - | ✓ | 198 | 33 | 47 | 23.74 |
| wbc | Healthcare | - | ✓ | 378 | 30 | 21 | 5.60 |
| arrhythmia | Healthcare | - | ✓ | 452 | 274 | 66 | 15.00 |
| Parkinson | Healthcare | - | ✓ | 195 | 22 | 147 | 75.38 |
| mnist | Image | - | ✓ | 7603 | 100 | 700 | 9.21 |
| optdigits | Image | - | ✓ | 5216 | 64 | 150 | 2.88 |
| pendigits | Image | - | ✓ | 6870 | 16 | 156 | 2.27 |
| imgseg | Image | ✓ | - | 2310 | 18 | 990 | 42.86 |
| fault | Physical&Chemistry | - | ✓ | 1941 | 27 | 673 | 34.67 |
| musk | Physical&Chemistry | - | ✓ | 3062 | 166 | 97 | 3.17 |
| wine | Physical&Chemistry | ✓ | - | 129 | 13 | 10 | 7.75 |
| census | Sociology | - | ✓ | 299285 | 500 | 18568 | 6.20 |
| comm.and.crime | Sociology | ✓ | - | 1994 | 101 | 993 | 49.80 |
| donors | Sociology | - | ✓ | 619326 | 10 | 36710 | 5.93 |

*Table 4.* The statistics of out-of-domain datasets.

| Dataset | Category | Train | Test | # Samples | # Features | # Anomaly | % Anomaly |
|---|---|:---:|:---:|---:|---:|---:|---:|
| **Out-of-domain datasets** | | | | | | | |
| cover | Botany | - | ✓ | 286048 | 10 | 2747 | 0.96 |
| Wilt | Botany | - | ✓ | 4819 | 5 | 257 | 5.33 |
| SpamBase | Document | - | ✓ | 4207 | 57 | 1679 | 39.91 |
| campaign | Finance | - | ✓ | 41188 | 62 | 4640 | 11.27 |
| fraud | Finance | - | ✓ | 284807 | 29 | 492 | 0.17 |
| glass | Forensic | - | ✓ | 214 | 7 | 9 | 4.21 |
| speech | Linguistics | - | ✓ | 3686 | 400 | 61 | 1.65 |
| vowels | Linguistics | - | ✓ | 1456 | 12 | 50 | 3.43 |
| backdoor | Network | - | ✓ | 95329 | 196 | 2329 | 2.44 |
| amazon | NLP | - | ✓ | 10000 | 768 | 500 | 5.00 |
| Ionosphere | Oryctognosy | - | ✓ | 351 | 32 | 126 | 35.90 |

# B. Algorithm Description of OFA-TAD

The algorithmic description of OFA-TAD is summarized in Algo. 1 and Algo. 2.

## C. Testing Time Complexity

We analyze the computational cost of OFA-TAD on a target dataset under the OFA protocol. Let $n$ denote the number of target training samples used as context, $m$ the number of target test samples to score, $d$ the feature dimension, $M$ the number of transformations (views), $K$ the number of retrieved neighbors per view, and $D$ the hidden dimension of each expert. In the testing phase, the time complexity consists of two main components: multi-view preprocessing and model inference.

For multi-view preprocessing, we first apply each transformation $\mathcal{T}_m$ to the context set and the test samples, which costs $\mathcal{O}(nd)$ for the context set and $\mathcal{O}(md)$ for the test set per view (constants depend on the transformer), yielding $\mathcal{O}(M(n+m)d)$ overall. We then perform $k$NN retrieval to obtain Top-$K$ neighbors for each test sample from the context set in every view. Under a standard exact $k$NN formulation based on brute-force distance computation, this retrieval step costs $\mathcal{O}(M \cdot mnd)$ and typically dominates the overall runtime.

The model inference further decomposes into view-wise embedding/score computation and gating-based fusion. Given retrieved distances of length $K$, each expert processes a $K$-token sequence with hidden size $D$, leading to $\mathcal{O}(M \cdot K \cdot D)$ per sample for producing view embeddings/scores, and the gating network adds $\mathcal{O}(MD)$ per sample. Therefore, scoring $m$ test samples costs $\mathcal{O}(m \cdot MKD)$, which is typically cheaper than neighborhood retrieval.

Overall, the testing-time complexity is dominated by neighbor retrieval and can be summarized as $\mathcal{O}(M \cdot mnd)$ (with lower-order $\mathcal{O}(M(n+m)d + m \cdot MKD)$ terms).

## D. Datasets and Split

Table 3 and Table 4 summarize the datasets used in our OFA evaluation. We report the number of samples, feature dimensionality, anomaly count and ratio, along with the semantic category through grouping. *Train/Test* indicates whether the dataset is used as a *source* dataset for one-time pre-training (Train ✓) or as a *target* dataset for OFA evaluation (Test ✓).

## E. Baselines under the OFA Protocol

We evaluate all baselines under the same dataset split and without using any target-domain labels for model fitting. For each target dataset, we provide the target training split as the only in-domain resource available at test time.

**Target-domain usage.** For classic and deep unsupervised baselines (e.g., iForest, LOF, KNN, AutoEncoder (AE), DeepSVDD (DSVDD), DRL, LUNAR, MCM, DisentAD), we follow their standard per-dataset evaluation protocol: the model (or its statistics) is fitted on the target training split and then evaluated on the target test split. In contrast, OFA-TAD is trained *once* on the source datasets and is directly transferred to every target without any target-specific retraining; it only uses the target training split as *context* for neighbor retrieval and distance normalization during inference.

**Hyperparameters and tuning.** OFA-TAD does not perform any per-dataset hyperparameter tuning and uses a single fixed set of hyperparameters across all target datasets. For OFO baselines, to avoid exhaustive tuning while still providing competitive configurations, we perform a random hyperparameter search within a predefined search space separately on each target dataset. This search also includes selecting the feature transformation used by each baseline, so each baseline is evaluated with its best-performing transformation per dataset.

**Thresholding for F1.** AUROC/AUPRC are threshold-free and are computed directly from anomaly scores. For F1, we follow the setting used in DRL: given anomaly scores, we choose a percentile-based decision threshold and then compute F1 from the resulting binary predictions. This yields a unified thresholding protocol for all methods.

## F. Additional Comparison Results

### F.1. Comparison in More Metrics

Table 5 reports AUPRC results. OFA-TAD achieves the best average AUPRC (0.6629), outperforming the strongest baseline on average (MCM, 0.6259) by 3.70 points. Meanwhile, the best-performing method varies substantially across datasets (e.g., MCM on *breastw* with 0.9977, DSVDD on *cardio* with 0.8425, and DRL on *cover* with 0.8231), indicating that no single baseline dominates all targets.

Table 6 reports F1 under the unified thresholding protocol. OFA-TAD again ranks first on average (0.6352), exceeding the

Table 5. AUPRC comparison between baselines and OFA-TAD. The best results are highlighted in **bold**.

| Dataset | LOF | KNN | iForest | DSVDD | AE | MCM | LUNAR | DRL | DisentAD | OFA-TAD |
|---|---|---|---|---|---|---|---|---|---|---|
| In-Domain Target Datasets | | | | | | | | | | |
| abalone | 0.8273 | 0.8799 | 0.8496 | 0.8056 | 0.8856 | 0.8602 | 0.8839 | 0.8862 | 0.8700 | **0.8903** |
| arrhythmia | 0.6114 | 0.6034 | 0.5769 | 0.5091 | 0.3999 | 0.5832 | 0.5677 | 0.5676 | 0.5396 | **0.6212** |
| breastw | 0.9532 | 0.9962 | 0.9973 | 0.9894 | 0.9670 | **0.9977** | 0.9689 | 0.9945 | 0.9961 | 0.9740 |
| cardio | 0.7599 | 0.7663 | 0.7886 | **0.8425** | 0.8194 | 0.5593 | 0.7826 | 0.8117 | 0.8085 | 0.8058 |
| census | 0.1175 | 0.1626 | 0.1451 | 0.1905 | 0.2074 | 0.1650 | 0.1654 | 0.1500 | **0.4003** | 0.1748 |
| donors | 0.7773 | 0.9725 | 0.4345 | 0.2695 | 0.4752 | 0.9293 | 0.8283 | 0.4157 | 0.6024 | **0.9929** |
| fault | 0.5044 | 0.6198 | 0.6002 | 0.5512 | 0.5953 | **0.6522** | 0.5725 | 0.6309 | 0.6259 | 0.6150 |
| Hepatitis | 0.4289 | 0.3063 | 0.4289 | **0.5444** | 0.3985 | 0.4515 | 0.3115 | 0.4344 | 0.4767 | 0.4846 |
| lympho | 0.5646 | 0.4628 | 0.9738 | **0.9867** | 0.4061 | 0.9246 | 0.8559 | 0.8784 | 0.5709 | 0.9182 |
| mammography | 0.3500 | 0.3893 | 0.4125 | 0.4317 | 0.3899 | 0.4303 | 0.2619 | 0.3813 | 0.4150 | **0.5395** |
| mnist | 0.8380 | 0.7693 | 0.5515 | 0.5701 | 0.6985 | 0.8632 | 0.6710 | **0.8658** | 0.3030 | 0.8104 |
| musk | **1.0000** | **1.0000** | 0.7720 | 0.8927 | **1.0000** | 0.9917 | 0.9279 | 0.9986 | 0.7808 | **1.0000** |
| optdigits | 0.8172 | 0.4883 | 0.1415 | 0.0591 | 0.1608 | 0.7977 | 0.5825 | 0.2555 | **0.8424** | 0.8162 |
| Parkinson | 0.9299 | 0.8202 | **0.9607** | 0.9261 | 0.9292 | 0.7927 | 0.8336 | 0.9211 | 0.9209 | 0.9348 |
| pendigits | 0.6549 | 0.7236 | 0.5019 | 0.1925 | 0.5329 | 0.7068 | 0.8508 | 0.4269 | 0.8996 | **0.9758** |
| pima | 0.6856 | 0.7181 | 0.7194 | 0.6707 | 0.7122 | 0.7062 | 0.6887 | **0.7210** | 0.6856 | 0.7037 |
| satimage-2 | 0.8846 | 0.9669 | 0.9461 | 0.8387 | **0.9779** | 0.9774 | 0.6179 | 0.8602 | 0.5736 | 0.9610 |
| shuttle | 0.9458 | 0.9225 | 0.9862 | 0.9622 | 0.9683 | 0.9683 | 0.8138 | 0.9693 | 0.9958 | **0.9961** |
| thyroid | 0.6055 | 0.8094 | 0.7506 | 0.8038 | 0.7264 | 0.7817 | 0.6778 | 0.7867 | **0.8685** | 0.8026 |
| wbc | 0.8573 | 0.8438 | 0.8317 | 0.7910 | 0.8023 | 0.8413 | 0.7638 | **0.9114** | 0.8120 | 0.8092 |
| WDBC | 0.9833 | 0.9573 | 0.9677 | 0.8723 | 0.9485 | 0.9462 | 0.9250 | 0.9843 | 0.9605 | **0.9933** |
| WPBC | 0.4119 | 0.4037 | 0.3799 | 0.3872 | 0.3848 | 0.4299 | 0.3984 | 0.4101 | **0.4314** | 0.4020 |
| yeast | 0.4866 | 0.4821 | 0.4697 | 0.4858 | 0.4799 | 0.4766 | **0.5504** | 0.5030 | 0.5125 | 0.4912 |
| Out-of-Domain Target Datasets | | | | | | | | | | |
| amazon | 0.1010 | 0.0999 | 0.0947 | 0.0952 | 0.0987 | 0.0974 | **0.1936** | 0.0961 | 0.1182 | 0.1028 |
| backdoor | 0.3751 | 0.4799 | 0.0943 | 0.1008 | 0.8558 | 0.6154 | 0.6293 | **0.8619** | 0.1230 | 0.7300 |
| campaign | 0.2768 | 0.4467 | 0.4614 | 0.4241 | 0.4614 | **0.5570** | 0.4241 | 0.4590 | 0.4399 | 0.4515 |
| cover | 0.1598 | 0.1045 | 0.0560 | 0.0595 | 0.0661 | 0.1510 | 0.4487 | **0.8231** | 0.7178 | 0.5370 |
| fraud | 0.0027 | 0.2535 | 0.2373 | 0.1819 | 0.2777 | 0.5306 | 0.3981 | 0.2309 | **0.6142** | 0.3870 |
| glass | 0.0952 | 0.0931 | 0.0956 | 0.0904 | 0.0946 | 0.1447 | 0.1010 | 0.1008 | **0.4890** | 0.1768 |
| ionosphere | 0.8607 | 0.9590 | 0.8559 | 0.9066 | 0.9695 | 0.9772 | 0.9700 | **0.9774** | 0.9765 | 0.9742 |
| SpamBase | 0.7271 | 0.8135 | 0.8760 | 0.7971 | 0.8200 | 0.7833 | 0.8160 | 0.8455 | 0.6524 | **0.8924** |
| speech | 0.0295 | 0.0273 | 0.0318 | 0.0290 | 0.0272 | 0.0324 | 0.0578 | 0.0472 | **0.0685** | 0.0368 |
| vowels | 0.3056 | 0.3021 | 0.1156 | 0.1052 | 0.3018 | 0.3668 | 0.3179 | 0.3175 | **0.5126** | 0.3200 |
| Wilt | 0.1574 | 0.1746 | 0.0881 | 0.0778 | 0.0825 | 0.1909 | **0.3168** | 0.2684 | 0.1876 | 0.2165 |
| Average AUPRC | 0.5614 | 0.5829 | 0.5351 | 0.5130 | 0.5565 | 0.6259 | 0.5933 | 0.6116 | 0.6115 | **0.6629** |

next best method (LUNAR, 0.6121) by 2.31 points, while several strong deep baselines cluster around 0.58–0.60 (MCM 0.6012, DRL 0.5921, DisentAD 0.5867). Consistent with the per-dataset entries, some datasets are threshold-sensitive and can favor different methods (e.g., DisentAD on *census* with 0.4044, DRL on *backdoor* with 0.8673, and LUNAR on *WPBC* with 0.5751). Crucially, OFA-TAD achieves the best average F1 without per-dataset hyperparameter tuning, while baselines rely on target-domain fitting for each dataset.

### F.2. F1 Ranking

Figure 8 visualizes the average ranking under F1. Consistent with the AUROC/AUPRC rank ladders in Figure 4, OFA-TAD stays among the top methods under this threshold-dependent metric, suggesting that its advantage is not tied to a particular metric or operating point. Meanwhile, several baselines exhibit larger rank shifts compared with their AUROC/AUPRC behavior, indicating metric-sensitive performance and less stable cross-dataset generalization.

## G. Robustness to Source Dataset Combinations

To verify the robustness of OFA-TAD to the specific choice of source datasets, we conduct a broader source-combination analysis. Using the same 7 source and 34 target datasets evaluated in the main text, we randomly sampled source subsets of

Table 6. F1 comparison between baselines and OFA-TAD. The best results are highlighted in **bold**.

| Dataset | LOF | KNN | iForest | DSVDD | AE | MCM | LUNAR | DRL | DisentAD | OFA-TAD |
|---|---|---|---|---|---|---|---|---|---|---|
| | | | | In-Domain Target Datasets | | | | | | |
| abalone | 0.7785 | 0.8107 | 0.7710 | 0.7395 | 0.8140 | 0.7720 | 0.8154 | 0.8131 | 0.7957 | **0.8229** |
| arrhythmia | 0.5758 | 0.5909 | 0.5909 | 0.5121 | 0.4848 | 0.5758 | 0.5780 | 0.5061 | 0.5152 | **0.6061** |
| breastw | 0.9456 | 0.9749 | **0.9816** | 0.9682 | 0.9414 | 0.9749 | 0.9526 | 0.9782 | 0.9707 | 0.9439 |
| cardio | 0.6705 | 0.6818 | 0.6708 | **0.7477** | 0.7273 | 0.5254 | 0.6955 | 0.7375 | 0.7364 | 0.7205 |
| census | 0.0592 | 0.1447 | 0.1112 | 0.2025 | 0.2228 | 0.1576 | 0.1620 | 0.1542 | **0.4044** | 0.1593 |
| donors | 0.8654 | 0.8334 | 0.4517 | 0.2665 | 0.5209 | 0.9554 | **0.9913** | 0.4224 | 0.6174 | 0.9722 |
| fault | 0.5067 | 0.5557 | 0.5441 | 0.5224 | 0.5750 | 0.5768 | 0.4878 | 0.5554 | **0.5813** | 0.5486 |
| Hepatitis | 0.3529 | 0.3077 | **0.4923** | 0.4769 | 0.3077 | 0.3286 | 0.4869 | 0.3538 | 0.4462 | 0.4000 |
| lympho | 0.6667 | 0.5000 | 0.8667 | **0.9667** | 0.0000 | 0.8667 | 0.7778 | 0.8333 | 0.5333 | 0.9000 |
| mammography | 0.4115 | 0.4077 | 0.4238 | 0.4669 | 0.4346 | 0.4414 | 0.3015 | 0.4000 | 0.4085 | **0.5200** |
| mnist | 0.7471 | 0.7000 | 0.5289 | 0.5577 | 0.6886 | **0.8014** | 0.6165 | 0.7969 | 0.3137 | 0.7423 |
| musk | **1.0000** | **1.0000** | 0.6977 | 0.8598 | **1.0000** | 0.9898 | 0.9630 | 0.9897 | 0.6742 | **1.0000** |
| optdigits | **0.8400** | 0.5552 | 0.1160 | 0.0080 | 0.0800 | 0.7573 | 0.6250 | 0.2800 | 0.8245 | 0.8227 |
| Parkinson | 0.8912 | 0.8571 | 0.8694 | 0.8830 | 0.8912 | 0.8490 | **0.9257** | 0.8816 | 0.9061 | 0.8830 |
| pendigits | 0.7244 | 0.7115 | 0.5308 | 0.2269 | 0.5513 | 0.6782 | 0.8260 | 0.4654 | 0.8372 | **0.9308** |
| pima | 0.6493 | 0.6493 | **0.6910** | 0.6560 | 0.6903 | 0.6731 | 0.6886 | 0.6836 | 0.6724 | 0.6709 |
| satimage-2 | 0.8169 | 0.9014 | 0.8926 | 0.7859 | **0.9437** | 0.9296 | 0.6645 | 0.8282 | 0.5606 | 0.9211 |
| shuttle | 0.9752 | 0.9718 | 0.9642 | 0.9589 | 0.9655 | 0.9793 | 0.6720 | 0.9817 | 0.9781 | **0.9855** |
| thyroid | 0.5269 | 0.7527 | 0.7978 | 0.7226 | 0.7097 | 0.7383 | 0.6690 | 0.7247 | **0.8022** | 0.7398 |
| wbc | 0.7619 | 0.7143 | 0.7048 | 0.6952 | 0.6667 | 0.7455 | 0.7006 | **0.8000** | 0.7333 | 0.7143 |
| WDBC | 0.9000 | 0.9000 | 0.8800 | 0.7600 | 0.8000 | 0.9091 | 0.8989 | 0.9400 | 0.9200 | **0.9600** |
| WPBC | 0.3617 | 0.3404 | 0.3574 | 0.3745 | 0.3404 | 0.3625 | **0.5751** | 0.3532 | 0.4255 | 0.3234 |
| yeast | 0.4872 | 0.4615 | 0.4458 | 0.4706 | 0.4813 | 0.4669 | **0.6799** | 0.4978 | 0.5037 | 0.4821 |
| | | | | Out-of-Domain Target Datasets | | | | | | |
| amazon | 0.0820 | 0.0920 | 0.0820 | 0.0892 | 0.0840 | 0.0756 | **0.1811** | 0.0900 | 0.1272 | 0.0964 |
| backdoor | 0.4063 | 0.5019 | 0.0078 | 0.1513 | 0.8518 | 0.7478 | 0.8072 | **0.8673** | 0.1561 | 0.8235 |
| campaign | 0.3039 | 0.4112 | 0.4406 | 0.4398 | 0.4782 | **0.5797** | 0.4231 | 0.4944 | 0.4352 | 0.4646 |
| cover | 0.2035 | 0.0797 | 0.0838 | 0.0376 | 0.0444 | 0.1359 | 0.5051 | **0.8100** | 0.7468 | 0.5761 |
| fraud | 0.0000 | 0.3557 | 0.3232 | 0.2659 | 0.3557 | 0.5769 | 0.4198 | 0.2691 | **0.6325** | 0.4871 |
| glass | 0.0000 | 0.0000 | 0.0222 | 0.0000 | 0.0000 | 0.0800 | 0.2357 | 0.0667 | **0.4000** | 0.1791 |
| ionosphere | 0.7012 | 0.8413 | 0.7238 | 0.8063 | 0.8968 | 0.9024 | 0.8889 | **0.9349** | 0.9238 | 0.9032 |
| SpamBase | 0.7397 | 0.7385 | 0.8011 | 0.7571 | 0.7933 | 0.7251 | 0.7643 | 0.8058 | 0.6310 | **0.8180** |
| speech | 0.0164 | 0.0164 | 0.0361 | 0.0164 | 0.0164 | 0.0387 | 0.0876 | 0.0721 | **0.1049** | 0.0426 |
| vowels | 0.3600 | 0.2600 | 0.1400 | 0.1160 | 0.2800 | 0.3880 | 0.3774 | 0.3520 | **0.5320** | 0.3120 |
| Wilt | 0.1673 | 0.0856 | 0.0202 | 0.0397 | 0.0195 | 0.1354 | 0.3685 | **0.3917** | 0.0965 | 0.1245 |
| Average F1 | 0.5440 | 0.5501 | 0.5018 | 0.4867 | 0.5193 | 0.6012 | 0.6121 | 0.5921 | 0.5867 | **0.6352** |

Table 7. Source-combination robustness over random trials (measured by AVG AUROC %).

| Source Size | Valid Trials | Mean AUROC | Std |
|---|---|---|---|
| 2 | 21 | 82.44 | 0.10 |
| 3 | 30 | 82.74 | 0.17 |
| 4 | 30 | 82.85 | 0.15 |
| 5 | 21 | 82.98 | 0.15 |

size 2 to 5, with 30 independent trials per size (we evaluated sizes 2 and 5 since $C(7, 2) = C(7, 5) = 21$).

The results across different source combinations are summarized in Table 7. The standard deviation of AUROC remains below 0.17% across all subset sizes, indicating that the transferability of OFA-TAD is highly stable and not reliant on a particular "lucky" source configuration. Furthermore, the mean AUROC monotonically improves from 82.44% (when using 2 source datasets) to 82.98% (when using 5 source datasets), confirming that incorporating more diverse source data reliably benefits cross-domain generalization.

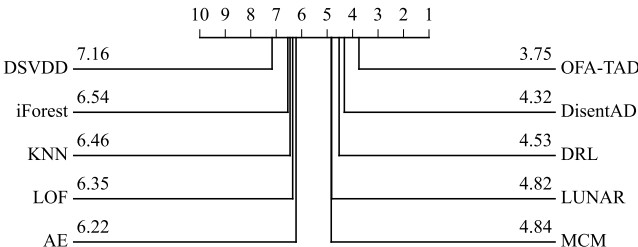

*Figure 8.* Average rank under F1 across target datasets (the lower the better).

## H. Full Ablation Results

Tables 8–10 summarize full ablation results across all target datasets under AUROC, AUPRC, and F1. Overall, OFA–TAD achieves the best average performance, while removing any component consistently degrades results, indicating that the proposed design choices are complementary. Among architectural variants, discarding attention pooling typically causes the largest drop, suggesting that adaptively emphasizing informative neighbor ranks is crucial when anomaly evidence is sparse, whereas removing gating or MoE mainly hurts datasets that exhibit stronger transformation sensitivity. For training variants, using the full mixture of pseudo-anomaly synthesis strategies is generally most robust, while removing any single strategy can harm specific datasets, reflecting the diversity of anomaly patterns.

## I. Extended Context-Size Study

Figure. 9 shows that increasing the amount of target-domain context generally improves performance, with the largest gains appearing when moving from very limited context to a moderate ratio. As the context size continues to grow, the curves often plateau, indicating diminishing returns once neighborhood statistics become sufficiently stable. In some cases, overly large context can slightly reduce performance, possibly because additional context samples introduce noisy or less relevant neighbors that dilute local density cues used by distance-based scoring. Meanwhile, sensitivity to context varies across datasets, reflecting heterogeneous neighborhood structures and anomaly separability under domain shift.

## J. Additional Gating Visualizations

The gating heatmaps in Figure. 10 further confirm that the reliability of distance views varies substantially across target datasets, with the model often concentrating its mass on one or two transformations rather than using all views uniformly. Notably, the preferred views differ across datasets, indicating that no single transformation is consistently optimal under domain shift. Overall, these results support that the learned gating mechanism enables sample- and dataset-adaptive view selection, which helps mitigate transformation sensitivity in OFA inference.

*Table 8.* Ablation results w.r.t. per-dataset AUROC. The best results are highlighted in **bold**, where G: gating, A: attention, P: position, N: noise injection, E: extrapolation, I: interpolation, M: masking.

| Dataset | OFA-TAD | w/o G | w/o MoE | w/o A | w/o P | w/o N | w/o E | w/o I | w/o M |
|---|---|---|---|---|---|---|---|---|---|
| *In-Domain Target Datasets* | | | | | | | | | |
| abalone | **0.8178** | 0.8165 | 0.8103 | 0.8144 | 0.8065 | 0.8028 | 0.8055 | 0.8078 | 0.8008 |
| arrhythmia | 0.8095 | **0.8114** | 0.8092 | 0.8038 | 0.8059 | 0.7980 | 0.7997 | 0.8078 | 0.7979 |
| breastw | 0.9791 | 0.9959 | **0.9964** | 0.9955 | 0.9950 | 0.9736 | 0.9627 | 0.9785 | 0.9827 |
| cardio | 0.9322 | 0.9536 | **0.9559** | 0.9548 | 0.9247 | 0.9152 | 0.9225 | 0.9358 | 0.9314 |
| census | 0.6955 | 0.6895 | 0.6923 | 0.6858 | 0.6964 | 0.6907 | 0.6892 | 0.6932 | **0.6993** |
| donors | 0.9997 | **0.9999** | 0.9999 | 0.9998 | 0.9989 | 0.9889 | 0.9901 | 0.9998 | 0.9995 |
| fault | 0.5762 | 0.6049 | **0.6067** | 0.6015 | 0.5862 | 0.5716 | 0.5709 | 0.5818 | 0.5782 |
| Hepatitis | 0.7353 | 0.7504 | 0.7557 | **0.7632** | 0.7290 | 0.7407 | 0.7330 | 0.7380 | 0.7276 |
| lympho | 0.9911 | 0.9914 | 0.9930 | 0.9804 | 0.9878 | 0.9549 | 0.9563 | **1.0000** | 1.0000 |
| mammography | 0.9000 | 0.8914 | 0.8917 | 0.8893 | 0.8867 | 0.8944 | 0.8972 | **0.9005** | 0.8957 |
| mnist | 0.9433 | 0.9434 | 0.9416 | 0.9400 | 0.9434 | 0.9398 | 0.9414 | **0.9449** | 0.9426 |
| musk | **1.0000** | 1.0000 | 1.0000 | 0.9993 | 0.9999 | 0.9811 | 0.9856 | 1.0000 | 0.9916 |
| optdigits | 0.9930 | 0.9850 | 0.9757 | 0.9830 | 0.9925 | **0.9938** | 0.9936 | 0.9935 | 0.9915 |
| Parkinson | **0.7164** | 0.6536 | 0.6706 | 0.6500 | 0.6366 | 0.6908 | 0.6936 | 0.6883 | 0.6956 |
| pendigits | 0.9990 | 0.9980 | 0.9961 | 0.9979 | **0.9994** | 0.9971 | 0.9969 | 0.9989 | 0.9991 |
| pima | 0.6959 | **0.7481** | 0.7471 | 0.7456 | 0.7056 | 0.6957 | 0.7088 | 0.7102 | 0.6872 |
| satimage-2 | 0.9964 | 0.9975 | 0.9973 | 0.9975 | **0.9977** | 0.9559 | 0.9389 | 0.9967 | 0.9961 |
| shuttle | **0.9998** | 0.9995 | 0.9995 | 0.9990 | 0.9996 | 0.9970 | 0.9989 | 0.9997 | 0.9998 |
| thyroid | 0.9809 | 0.9818 | **0.9821** | 0.9814 | 0.9767 | 0.9717 | 0.9706 | 0.9811 | 0.9810 |
| wbc | 0.9516 | **0.9679** | 0.9667 | 0.9675 | 0.9523 | 0.9482 | 0.9476 | 0.9538 | 0.9468 |
| WDBC | 0.9996 | **1.0000** | 1.0000 | 0.9996 | 0.9991 | 0.9947 | 0.9963 | 0.9996 | 0.9975 |
| WPBC | 0.4830 | 0.4855 | 0.4840 | **0.4899** | 0.4758 | 0.4805 | 0.4858 | 0.4789 | 0.4843 |
| yeast | 0.4656 | 0.4378 | 0.4380 | 0.4301 | 0.4616 | **0.4827** | 0.4617 | 0.4602 | 0.4595 |
| *Out-of-Domain Target Datasets* | | | | | | | | | |
| amazon | 0.5469 | 0.5312 | 0.5289 | 0.5299 | 0.5446 | 0.5473 | 0.5463 | 0.5444 | **0.5478** |
| backdoor | **0.9592** | 0.9441 | 0.9446 | 0.9424 | 0.9541 | 0.9583 | 0.9564 | 0.9576 | 0.9576 |
| campaign | 0.7564 | 0.7425 | 0.7440 | 0.7388 | 0.7556 | 0.7573 | **0.7575** | 0.7565 | 0.7529 |
| cover | **0.9627** | 0.9542 | 0.9540 | 0.9433 | 0.9319 | 0.8854 | 0.8698 | 0.9584 | 0.9423 |
| fraud | 0.8785 | 0.9304 | 0.9312 | **0.9321** | 0.9039 | 0.7847 | 0.7717 | 0.8833 | 0.8828 |
| glass | **0.6690** | 0.5969 | 0.5890 | 0.6127 | 0.5989 | 0.6153 | 0.6247 | 0.6096 | 0.6142 |
| ionosphere | 0.9639 | 0.9228 | 0.9036 | 0.9144 | **0.9664** | 0.9618 | 0.9656 | 0.9655 | 0.9632 |
| SpamBase | 0.8599 | 0.8612 | **0.8662** | 0.8371 | 0.8493 | 0.8414 | 0.8441 | 0.8600 | 0.8564 |
| speech | 0.4891 | 0.3707 | 0.3677 | 0.3689 | 0.4814 | 0.4791 | 0.4845 | 0.4804 | **0.5007** |
| vowels | **0.8151** | 0.7833 | 0.7529 | 0.7804 | 0.8115 | 0.7931 | 0.7788 | 0.8131 | 0.8145 |
| Wilt | 0.8102 | 0.5998 | 0.6009 | 0.5658 | 0.8016 | 0.8076 | 0.8005 | 0.7985 | **0.8141** |
| Average AUROC | **0.8345** | 0.8218 | 0.8204 | 0.8187 | 0.8281 | 0.8203 | 0.8190 | 0.8317 | 0.8304 |

*Table 9.* Ablation results w.r.t. per-dataset AUPRC. The best results are highlighted in **bold**, where G: gating, A: attention, P: position, N: noise injection, E: extrapolation, I: interpolation, M: masking.

| Dataset | OFA-TAD | w/o G | w/o MoE | w/o A | w/o P | w/o N | w/o E | w/o I | w/o M |
|---|---|---|---|---|---|---|---|---|---|
| *In-Domain Target Datasets* | | | | | | | | | |
| abalone | 0.8903 | **0.8905** | 0.8879 | 0.8895 | 0.8822 | 0.8763 | 0.8760 | 0.8827 | 0.8791 |
| arrhythmia | 0.6212 | **0.6298** | 0.6294 | 0.5856 | 0.6088 | 0.5712 | 0.5706 | 0.6153 | 0.5846 |
| breastw | 0.9740 | 0.9958 | **0.9964** | 0.9951 | 0.9941 | 0.9480 | 0.9346 | 0.9743 | 0.9774 |
| cardio | 0.8058 | 0.8358 | **0.8436** | 0.8413 | 0.7950 | 0.7006 | 0.7151 | 0.8097 | 0.8075 |
| census | 0.1748 | 0.1703 | 0.1724 | 0.1688 | 0.1757 | 0.1696 | 0.1694 | 0.1731 | **0.1808** |
| donors | 0.9929 | 0.9980 | **0.9981** | 0.9967 | 0.9698 | 0.8259 | 0.8715 | 0.9955 | 0.9872 |
| fault | 0.6150 | 0.6231 | 0.6242 | 0.6199 | **0.6274** | 0.5988 | 0.5966 | 0.6188 | 0.6224 |
| Hepatitis | 0.4846 | 0.4446 | 0.4546 | 0.4651 | 0.4597 | **0.5036** | 0.4745 | 0.4713 | 0.4857 |
| lympho | 0.9182 | 0.9193 | 0.9306 | 0.7926 | 0.8624 | 0.5558 | 0.6111 | **1.0000** | 1.0000 |
| mammography | **0.5395** | 0.5087 | 0.5079 | 0.4945 | 0.4506 | 0.4852 | 0.4776 | 0.5339 | 0.5129 |
| mnist | 0.8104 | 0.7919 | 0.7816 | 0.7789 | 0.7981 | 0.7809 | 0.7797 | **0.8124** | 0.8081 |
| musk | **1.0000** | 1.0000 | 1.0000 | 0.9861 | 0.9980 | 0.8401 | 0.8790 | 1.0000 | 0.9151 |
| optdigits | 0.8162 | 0.6956 | 0.5728 | 0.6708 | 0.8155 | **0.8389** | 0.8349 | 0.8220 | 0.7745 |
| Parkinson | **0.9348** | 0.9073 | 0.9144 | 0.9079 | 0.8957 | 0.9205 | 0.9197 | 0.9255 | 0.9287 |
| pendigits | 0.9758 | 0.9662 | 0.9405 | 0.9662 | **0.9868** | 0.9044 | 0.8987 | 0.9763 | 0.9780 |
| pima | 0.7037 | **0.7235** | 0.7209 | 0.7195 | 0.7149 | 0.7041 | 0.7088 | 0.7098 | 0.7009 |
| satimage-2 | 0.9610 | 0.9691 | **0.9692** | 0.9691 | 0.9652 | 0.7848 | 0.7805 | 0.9623 | 0.9593 |
| shuttle | **0.9961** | 0.9907 | 0.9882 | 0.9900 | 0.9895 | 0.9343 | 0.9722 | 0.9952 | 0.9954 |
| thyroid | 0.8026 | 0.7824 | 0.7858 | 0.7708 | 0.7197 | 0.6149 | 0.6166 | **0.8042** | 0.7961 |
| wbc | 0.8092 | **0.8387** | 0.8341 | 0.8381 | 0.8047 | 0.7454 | 0.7413 | 0.8114 | 0.8038 |
| WDBC | 0.9933 | **1.0000** | 1.0000 | 0.9944 | 0.9846 | 0.8709 | 0.8898 | 0.9933 | 0.9325 |
| WPBC | 0.4020 | 0.4042 | 0.4034 | **0.4073** | 0.4003 | 0.3998 | 0.4010 | 0.4010 | 0.3993 |
| yeast | 0.4912 | 0.4752 | 0.4750 | 0.4717 | 0.4894 | **0.4985** | 0.4897 | 0.4897 | 0.4889 |
| *Out-of-Domain Target Datasets* | | | | | | | | | |
| amazon | 0.1028 | 0.0987 | 0.0982 | 0.0985 | 0.1023 | 0.1037 | **0.1038** | 0.1023 | 0.1031 |
| backdoor | **0.7300** | 0.6983 | 0.6876 | 0.6771 | 0.6777 | 0.6942 | 0.7177 | 0.7158 | 0.6689 |
| campaign | 0.4515 | 0.4554 | **0.4592** | 0.4547 | 0.4535 | 0.4469 | 0.4435 | 0.4490 | 0.4510 |
| cover | **0.5370** | 0.4501 | 0.4739 | 0.3646 | 0.2013 | 0.3154 | 0.3442 | 0.4919 | 0.3834 |
| fraud | 0.3870 | 0.4266 | 0.4166 | 0.4311 | **0.4315** | 0.3186 | 0.3094 | 0.3954 | 0.3654 |
| glass | **0.1768** | 0.1211 | 0.1052 | 0.1230 | 0.1165 | 0.1330 | 0.1319 | 0.1279 | 0.1250 |
| ionosphere | 0.9742 | 0.9403 | 0.9231 | 0.9336 | **0.9755** | 0.9737 | 0.9751 | 0.9748 | 0.9738 |
| SpamBase | 0.8924 | 0.8966 | **0.9021** | 0.8620 | 0.8760 | 0.8475 | 0.8514 | 0.8909 | 0.8817 |
| speech | 0.0368 | 0.0300 | 0.0295 | 0.0300 | 0.0365 | 0.0355 | 0.0366 | 0.0364 | **0.0377** |
| vowels | **0.3200** | 0.3021 | 0.2836 | 0.3024 | 0.3080 | 0.2809 | 0.2725 | 0.3176 | 0.3150 |
| Wilt | 0.2165 | 0.1120 | 0.1123 | 0.1069 | 0.2084 | 0.2164 | 0.2114 | 0.2083 | **0.2218** |
| Average AUPRC | **0.6629** | 0.6498 | 0.6448 | 0.6383 | 0.6404 | 0.6011 | 0.6061 | 0.6614 | 0.6484 |

*Table 10.* Ablation results w.r.t. per-dataset F1. The best results are highlighted in **bold**, where G: gating, A: attention, P: position, N: noise injection, E: extrapolation, I: interpolation, M: masking.

| Dataset | OFA-TAD | w/o G | w/o MoE | w/o A | w/o P | w/o N | w/o E | w/o I | w/o M |
|---|---|---|---|---|---|---|---|---|---|
| In-Domain Target Datasets | | | | | | | | | |
| abalone | **0.8229** | 0.8214 | 0.8155 | 0.8188 | 0.8111 | 0.8085 | 0.8128 | 0.8136 | 0.8082 |
| arrhythmia | 0.6061 | **0.6263** | 0.6212 | 0.6061 | 0.5788 | 0.5606 | 0.5788 | 0.5879 | 0.5697 |
| breastw | 0.9439 | **0.9749** | 0.9749 | 0.9735 | 0.9741 | 0.9540 | 0.9372 | 0.9397 | 0.9573 |
| cardio | 0.7205 | 0.7424 | 0.7443 | **0.7462** | 0.7080 | 0.6875 | 0.7034 | 0.7284 | 0.7239 |
| census | 0.1593 | 0.1509 | 0.1545 | 0.1506 | 0.1591 | 0.1445 | 0.1453 | 0.1564 | **0.1722** |
| donors | 0.9722 | 0.9966 | **0.9971** | 0.9940 | 0.9635 | 0.9088 | 0.9173 | 0.9908 | 0.9837 |
| fault | 0.5486 | **0.5844** | 0.5840 | 0.5711 | 0.5501 | 0.5498 | 0.5480 | 0.5536 | 0.5471 |
| Hepatitis | 0.4000 | **0.4615** | 0.4615 | 0.4615 | 0.4615 | 0.4154 | 0.4154 | 0.4308 | 0.4154 |
| lympho | 0.9000 | 0.7778 | 0.8333 | 0.7222 | 0.8333 | 0.4667 | 0.4333 | **1.0000** | 1.0000 |
| mammography | **0.5200** | 0.4910 | 0.4923 | 0.4808 | 0.4592 | 0.4662 | 0.4692 | 0.5200 | 0.4992 |
| mnist | 0.7423 | 0.7152 | 0.7143 | 0.7048 | 0.7317 | 0.7326 | 0.7291 | **0.7449** | 0.7434 |
| musk | **1.0000** | 1.0000 | 1.0000 | 0.6797 | 0.8000 | 0.6000 | 0.6000 | 1.0000 | 0.6000 |
| optdigits | 0.8227 | 0.7067 | 0.6267 | 0.6956 | 0.8139 | **0.8573** | 0.8533 | 0.8347 | 0.8013 |
| Parkinson | 0.8830 | 0.8776 | 0.8776 | 0.8776 | **0.8871** | 0.8857 | 0.8857 | 0.8816 | 0.8844 |
| pendigits | 0.9308 | 0.8996 | 0.8526 | 0.9017 | **0.9487** | 0.8936 | 0.8885 | 0.9308 | 0.9372 |
| pima | 0.6709 | 0.7015 | 0.7015 | **0.7065** | 0.6754 | 0.6649 | 0.6784 | 0.6769 | 0.6597 |
| satimage-2 | 0.9211 | **0.9296** | 0.9296 | 0.9249 | 0.9211 | 0.7380 | 0.7352 | 0.9211 | 0.9183 |
| shuttle | 0.9855 | 0.9862 | 0.9860 | **0.9864** | 0.9863 | 0.9765 | 0.9831 | 0.9859 | 0.9858 |
| thyroid | 0.7398 | 0.7133 | 0.7097 | 0.7097 | 0.6839 | 0.6065 | 0.5763 | 0.7398 | **0.7484** |
| wbc | 0.7143 | **0.7302** | 0.7143 | 0.7302 | 0.7238 | 0.7143 | 0.7238 | 0.7143 | 0.7238 |
| WDBC | 0.9600 | **1.0000** | 1.0000 | 0.9667 | 0.9400 | 0.9000 | 0.9000 | 0.9600 | 0.9400 |
| WPBC | 0.3234 | 0.3262 | 0.3191 | **0.3404** | 0.3149 | 0.3191 | 0.3319 | 0.3191 | 0.3234 |
| yeast | 0.4821 | 0.4642 | 0.4615 | 0.4583 | 0.4809 | **0.4951** | 0.4806 | 0.4813 | 0.4777 |
| Out-of-Domain Target Datasets | | | | | | | | | |
| amazon | 0.0964 | 0.0867 | 0.0840 | 0.0860 | 0.0952 | 0.0980 | **0.1004** | 0.0960 | 0.0980 |
| backdoor | 0.8235 | **0.8396** | 0.8197 | 0.8069 | 0.8016 | 0.8027 | 0.7666 | 0.8116 | 0.7541 |
| campaign | 0.4646 | 0.4552 | 0.4558 | 0.4545 | 0.4626 | **0.4659** | 0.4637 | 0.4608 | 0.4653 |
| cover | **0.5761** | 0.4883 | 0.5075 | 0.4242 | 0.2533 | 0.2879 | 0.3421 | 0.5228 | 0.4059 |
| fraud | 0.4871 | **0.5198** | 0.5142 | 0.5156 | 0.5119 | 0.3896 | 0.3823 | 0.4901 | 0.4661 |
| glass | **0.1791** | 0.1111 | 0.1111 | 0.0741 | 0.1165 | 0.1156 | 0.0889 | 0.1111 | 0.1111 |
| ionosphere | 0.9032 | 0.8122 | 0.7857 | 0.7989 | **0.9048** | 0.9016 | 0.9000 | 0.9032 | 0.9016 |
| SpamBase | 0.8180 | 0.8124 | 0.8207 | 0.7955 | **0.8213** | 0.8096 | 0.8141 | 0.8191 | 0.8206 |
| speech | 0.0426 | 0.0328 | 0.0328 | 0.0328 | 0.0361 | 0.0393 | 0.0393 | 0.0393 | **0.0459** |
| vowels | **0.3120** | 0.2667 | 0.2800 | 0.2867 | 0.2880 | 0.2760 | 0.2800 | 0.3040 | 0.3040 |
| Wilt | 0.1245 | 0.0156 | 0.0195 | 0.0169 | 0.1222 | 0.1486 | **0.1510** | 0.1183 | 0.1323 |
| Average F1 | **0.6352** | 0.6211 | 0.6177 | 0.6029 | 0.6124 | 0.5788 | 0.5781 | 0.6349 | 0.6154 |

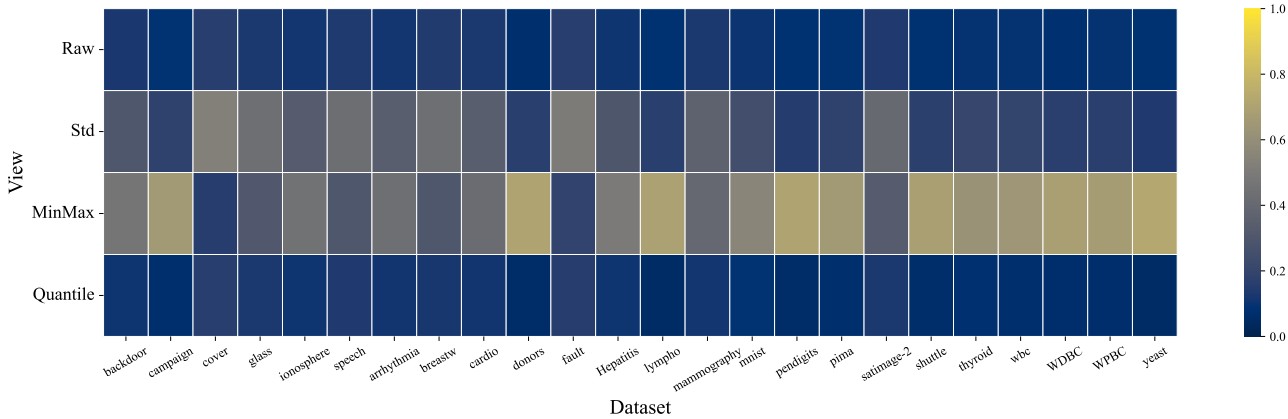

*Figure 9.* Performance with varying context size on the rest of target datasets.

*Figure 10.* Gating weights on additional target datasets.

