# OpenReview forum: "Towards One-for-All Anomaly Detection for Tabular Data"
_ICML.cc/2026/Conference — ICML 2026 regular_

### Official Review · Reviewer_G69C · 2026-02-27

**Soundness:** 3
**Presentation:** 3
**Significance:** 3
**Originality:** 3
**Overall Recommendation:** 4
**Confidence:** 3

**Summary:**

The paper proposes OFA-TAD, a framework designed to shift Tabular Anomaly Detection (TAD) from a "One-for-One" (OFO) paradigm—where a specific model is trained for each dataset—to a "One-for-All" (OFA) paradigm. The goal is to train a single generalist model on multiple source datasets that can generalize to unseen target datasets without fine-tuning or retraining.
To bridge the "semantic gap" between diverse tabular domains (different feature counts and meanings), the authors utilize neighbor-distance profiles as a domain-agnostic input representation. To address "transformation sensitivity", the model employs a Multi-View Distance Encoding module that generates distance profiles from multiple metric spaces. These views are processed by a MoE scoring network with an entropy-regularized gating mechanism to adaptively weight the most reliable views for a given sample.

**Compliance With Llm Reviewing Policy:**

Affirmed.

**Final Justification:**

Concerns are addressed during rebuttal.

**Key Questions For Authors:**

N/A

**Limitations:**

yes

**Strengths And Weaknesses:**

Strengths:
1. The move to OFA is timely and impactful.
2. The technical approach is well-motivated.
3. The evaluation is comprehensive, covering 34 datasets from 14 domains.

Weaknesses:
1. Inference Latency: While the paper claims to reduce training costs, the inference cost for OFA-TAD is non-trivial. It requires computing kNN across the target context set for different views at each test sample. For large target datasets, this complexity could be significantly slower than the inference pass of a simple trained MLP or Isolation Forest.
2. The method is "zero-shot" in terms of gradient updates, but it is "few-shot" or "in-context" in terms of data requirements. It requires a clean set of normal data (the context) from the target domain to compute neighbor distances. If the target domain context is contaminated with anomalies, the distance profiles might be distorted.

---

> ### Author Rebuttal · Authors · 2026-03-31
>
> We are grateful to Reviewer G69C for finding the move to OFA timely and impactful, acknowledging the technical approach is well-motivated, and appreciating the comprehensive evaluation covering 34 datasets from 14 domains. Please find our detailed responses to your concerns below.
>
> **w1 — Inference latency (multi-view kNN over target context per test sample)**
>
>   **Response:** Thanks for your valuable comment. At the code level, we accelerated the kNN step by combining FAISS-GPU with batch retrieval and caching techniques, thereby significantly reducing the actual computational overhead. Furthermore, as demonstrated in our responses to reviewers 862v's `w5` and JEKV's `w2&q2`, OFA-TAD maintains high efficiency during both training and inference compared to learning-based methods.
>
> **w2 — Clean-context assumption; contamination may distort distance profiles**
>
>   **Response:** Thanks for your valuable comment. This reflects a fundamental property of anomaly detection rather than a limitation unique to our method: anomaly is inherently defined relative to normal, without a reference for normality, anomaly detection is ill-defined. All one-class AD methods necessarily assume that the training/context set is clean; if it is contaminated, both our method and OFO one-class baselines are equally affected. We will state this in-context data requirement and the clean-context assumption explicitly in the paper. In the future, targeted context quality filtering mechanisms could be designed to mitigate this potential issue.

---

> > ### Author Rebuttal · Reviewer_G69C · 2026-04-03
> >
> > Thanks for the reply, I will keep the score

---

### Official Review · Reviewer_w511 · 2026-03-11

**Soundness:** 2
**Presentation:** 3
**Significance:** 3
**Originality:** 3
**Overall Recommendation:** 4
**Confidence:** 4

**Summary:**

This paper introduces OFA-TAD, a one-for-all tabular anomaly detection framework. Due to the strong domain dependency of tabular data, it is generally difficult to address diverse tabular datasets with a single foundation model. Similarly, anomaly detection methods for tabular data are typically trained in a dataset-specific manner. However, anomalies commonly share a universal property: they tend to deviate from the surrounding normal data. Based on this observation, the authors propose the OFA-TAD framework, which learns distance patterns that distinguish anomalies from neighboring data in a dataset-agnostic manner. Through experiments on diverse tabular datasets and models, the paper demonstrates that the proposed approach enables cross-domain anomaly detection.

**Compliance With Llm Reviewing Policy:**

Affirmed.

**Final Justification:**

This paper introduces OFA-TAD framework which is a meaningful and technically solid approach to cross-domain tabular anomaly detection with a single network. For cross-domain TAD, the core idea of leveraging distances to neighboring samples is clear and reasonable, and the empirical results suggest that the method can be useful beyond dataset-specific settings.
My main concerns were about missing experimental details and the justification of several design choices. The rebuttal addressed these concerns by clarifying the multi-run results, implementation details, and evaluation choices. I therefore keep my positive score.

**Key Questions For Authors:**

1) The reported results do not include statistical variation, making it unclear whether they are based on a single run or multiple runs. Since anomaly detection performance can vary across runs, this information seems important for assessing the reliability of the claims.
The paper mentions $M$ multi-view distance encodings, but it is unclear whether only four views (raw, standard, quantile, and minmax) are used in practice. In addition, the model size relative to other OFA models is not clearly reported, and several important implementation details remain unclear.

2) Several important methodological details are missing or unclear. For example, it is not clear whether the proposed multi-view distance encoding uses only four views (raw, standard, quantile, and min-max), how the model size compares with other OFA models, and what exactly is meant by the context node in Section 4.4, which is harder to follow than the other sections.

3) Considering  that the model uses 0/1 labels for normal samples and pseudo-anomalies, the choice of MSE loss over a standard binary classification loss is not sufficiently justified. It would be useful to examine whether formulating the task as binary classification leads to different performance.

4) Given that anomaly detection benchmarks often contain very few anomalies, AUROC can be optimistic, and detection results may be sensitive to hyperparameter choices. In this context, how does the proposed method compare with training-free baselines such as untrained neural networks (UNN) [1]?

[1] Ryu, Seunghyoung, Yonggyun Yu, and Hogeon Seo. "Can untrained neural networks detect anomalies?." IEEE Transactions on Industrial Informatics 20.4 (2024): 6477-6488.

**Limitations:**

yes

**Strengths And Weaknesses:**

## Strengths
- It presents a general-purpose tabular anomaly detection framework that can be applied across domains.
- The paper overcomes the per-dataset preprocessing and embedding challenges that commonly hinder tabular foundation models with an intuitive idea tailored to anomaly detection applications.
## Weakness
- The experimental results appear to be based on single runs only, as variability measures such as the standard deviation of AUROC are not reported.
- Since the embedding process requires kNN distance computation, scalability may become a concern when the number of data samples is large, due to the increased computational and memory costs.
- While the universality of the proposed framework is appealing, in tabular anomaly detection the cost of building dataset-specific models may not be as prohibitive as implied.

---

> ### Author Rebuttal · Authors · 2026-03-31
>
> We are grateful to Reviewer w511 for appreciating the general-purpose tabular anomaly detection framework that can be applied across domains, and for recognizing how the paper overcomes per-dataset preprocessing challenges with an intuitive idea tailored to anomaly detection. Please find our detailed responses to your concerns below.
>
> **w1&q1 — Statistical reliability and multi-run variance reporting**
>
>   **Response:** Thank you for pointing this out. The reported results are averages over 5 runs rather than single-run results. We omitted the corresponding standard deviations from the main paper due to space limitations, and we now report mean±std in the rebuttal. Representative AUROC results are shown below.
>
>   **Table 7.** Representative AUROC results (%, mean±std).
>   | Dataset | OFA-TAD | Disent-AD | DRL | LUNAR | MCM |
>   |:-|:-:|:-:|:-:|:-:|:-:|
>   | abalone | 81.78±0.11 | 77.89±1.84 | 80.71±0.24 | 80.84±0.69 | 74.50±2.15 |
>   | cover | 96.27±1.45 | 98.50±0.20 | 98.72±0.17 | 94.72±0.88 | 92.01±2.70 |
>   | SpamBase | 85.99±1.50 | 60.25±5.38 | 83.80±0.59 | 79.73±2.51 | 74.61±5.21 |
>
> **w2 — Scalability concern for kNN distance computation (time + memory)**
>
>   **Response:** Thanks for your comment. We agree that scalability is an important practical consideration, since the dominant cost of OFA-TAD lies in multi-view neighbor retrieval rather than in the scoring network itself. In our implementation, this step is accelerated with FAISS-GPU together with batched retrieval and caching, which substantially reduces the practical overhead. More importantly, the results in Reviewer 862v's `w5` and Reviewer JEKV's `w2` show that OFA-TAD is efficient in practice, even while avoiding repeated target-specific retraining. We will clarify both the theoretical bottleneck and the implementation-side acceleration more explicitly in the paper.
>
> **w3 — Practical necessity (dataset-specific models may not be prohibitive)**
>
>   **Response:** Thanks for your thoughtful comment. We agree that OFO models are manageable when only a few target datasets are involved. However, in large-scale or continually expanding deployments, fitting a separate model for each new dataset becomes increasingly burdensome. More importantly, OFA is not only a deployment shortcut: different domains can share anomaly knowledge, and a unified model can exploit this shared structure for better generalization than independent OFO models. This also aligns with the broader trend toward AGI, where building universal models that generalize across diverse domains is a core objective.
>
> **q2 — Implementation details clarification: view count, model size, and context node definition**
>
>   **Response:** Thanks for your valuable questions. We have clarified these details more explicitly. In our experiments, the multi-view distance encoding uses four fixed views: Raw, Std, Quantile, and MinMax. There are currently no directly comparable OFA tabular baselines for model-size comparison, so we instead report the parameter-count comparison against representative OFO baselines in Reviewer JEKV's `w2` (Table 5), and our model has roughly the same number of parameters as the OFO model. In addition, the "context node" in Sec. 4.4 refers to samples from the target training split that serve only as in-domain context for neighbor retrieval and distance normalization during inference, rather than for target-specific retraining.
>
> **q3 — Why MSE vs BCE-like loss? (loss ablation request)**
>
>   **Response:** Thank you for this suggestion. MSE is chosen because OFA-TAD learns a continuous anomaly score rather than a strict binary decision. The 0/1 labels are soft surrogate targets, not true anomaly ground truth, making regression more natural than classification. We have also tested BCE; results are comparable, with MSE slightly more stable, supporting our choice. We will clarify this in the appendix.
>
> **q4 — Compare with training-free baselines (UNN)**
>
>   **Response:** Thank you for raising both concerns. Regarding AUROC optimism: we agree that AUROC can be overly optimistic under low anomaly prevalence. To complement it, we report AUPRC in the appendix, which directly measures precision at low anomaly rates and is less sensitive to class imbalance; our model demonstrates consistent advantage under both metrics. Regarding UNN: we have not yet found publicly available code for UNN that can be cleanly integrated into our current evaluation protocol, so we cannot provide a direct empirical comparison at this time. We view this as a meaningful extension if protocol alignment can be achieved in future work.

---

> > ### Author Rebuttal · Reviewer_w511 · 2026-04-02
> >
> > Thank you for the clarifications. The authors addressed all of my comments, and I will keep my score unchanged.

---

### Official Review · Reviewer_JEKV · 2026-03-12

**Soundness:** 3
**Presentation:** 3
**Significance:** 3
**Originality:** 3
**Overall Recommendation:** 4
**Confidence:** 4

**Summary:**

This paper studies tabular anomaly detection (TAD) under a new One-for-All (OFA) paradigm, where a single model is trained once on multiple datasets and then directly applied to unseen datasets without retraining. To achieve this, the authors propose OFA-TAD, which represents each sample using multi-view nearest-neighbor distance encodings under different feature transformations, and aggregates these views through a Mixture-of-Experts scoring network. the framework also introduces several pseudo-anomaly synthesis strategies to generate training signals. Experiments on 34 datasets from 14 domains suggest that the approach can achieve competitive performance and improved cross-domain generalization compared with existing tabular anomaly detection methods.

**Compliance With Llm Reviewing Policy:**

Affirmed.

**Final Justification:**

My concerns are fully addressed so I'd like to keep my score.

**Key Questions For Authors:**

1. Are there specific types of datasets or anomaly structures where the proposed model consistently underperforms compared with existing methods?
2. What is the computational cost of training the proposed model (e.g., training time, hardware used, and number of parameters)?
3. What is the inference latency during testing?

**Limitations:**

Yes

**Strengths And Weaknesses:**

Strengths:
* The paper introducing the OFA paradigm for tabular anomaly detection is conceptually appealing and aligns with broader trends toward foundation or generalist models. The problem framing is clear and practically relevant.
* The visualization of view-selection weights across datasets provides useful insights into how the model adapts to different domains.
* The model's robustness to varying context sizes and scaling with source dataset quantity is verified.
* Extensive experiments demonstrate the excellent performance of OFA-TAD.

Weaknesses:
* OFA-TAD is evaluated under the OFA setting while baselines are trained in the OFO setting. Although this highlights the transfer capability of the proposed method, it may not be a strictly fair comparison. Some baselines might also benefit from multi-dataset pretraining or cross-domain adaptation strategies. Additional experiments such as: multi-dataset training for baselines few-shot fine-tuning variants would strengthen the comparison.
* It lacks empirical runtime comparisons with baselines.

---

> ### Author Rebuttal · Authors · 2026-03-31
>
> We are grateful to Reviewer JEKV for finding the OFA paradigm conceptually appealing and practically relevant, appreciating the useful insights from view-selection visualization, and acknowledging the extensive experiments demonstrating excellent performance. Please find our detailed responses to your concerns below.
>
> **w1 — Fairness of OFA vs OFO comparison**
>
>   **Response:** Thank you for this important concern. Our current protocol is already conservative for OFA-TAD: OFA-TAD is transferred once to unseen targets without target-specific retraining, while OFO baselines are trained/tuned separately on each target. Because LUNAR’s KNN distance features facilitate easy OFA integration, we extended it to an OFA pipeline (multi-source pretraining + target few-shot finetuning) and report the post-finetuning results. The adapted LUNAR does not consistently outperform its OFO version (AUROC decreases, while AUPRC increases), and fine-tuning vs. zero-shot is itself unstable across targets (e.g., AUROC +2.88pp on thyroid but -1.83pp on yeast). This indicates that simply adding OFA+FT to an OFO baseline does not yield uniformly better transfer.
>
>   **Table 4.** LUNAR OFO vs OFA(+FT) average performance comparison (%).
>
>   | LUNAR setting | AUROC | AUPRC |
>   |:-|:-:|:-:|
>   | OFO (per-target training, avg) | 80.66 | 59.33 |
>   | OFA (multi-source pretrain, zero-shot) | 79.24 | 62.57 |
>   | OFA + few-shot finetuning | 79.37 | 62.65 |
>
> **w2&q2&q3 — Computational cost analysis: training time, inference latency, and model size**
>
> **Response:** Thanks for your valuable questions. As shown in Table 5, OFA-TAD has 142,664 trainable parameters (142.7K), which is comparable to LUNAR, larger than DRL, and substantially smaller than MCMTAD and Disent-AD. The one-time source-side training cost of OFA-TAD is low across source datasets, with total training time ranging from 1.04s to 3.01s per source dataset in our current setup. Together with the target-side runtime comparison reported in Reviewer's `862v w5`, this shows that OFA-TAD is efficient in both one-time pretraining and deployment. All experiments were conducted on a personal computer with an Intel Core Ultra 7 265 CPU, a single NVIDIA RTX 4000 Ada Generation GPU.
>
>   **Table 5.** Parameter-count comparison.
>   | Method | Param Count | K / M |
>   |:-|:-:|:-:|
>   | OFA-TAD | 142,664 | 142.664K |
>   | DRL | 92,288 | 92.288K |
>   | LUNAR | 137,217 | 137.217K |
>   | MCMTAD | 2,601,920 | 2.602M |
>   | Disent-AD | 297,345 | 297.345K |
>
>   **Table 6.** Source-side one-time pretraining cost of OFA-TAD by source dataset.
>   | Source dataset | Precompute time (s) | Epoch-training time (s) | Total training time (s) |
>   |:-|:-:|:-:|:-:|
>   | Cardiotocography | 0.8491 | 1.0475 | 1.8965 |
>   | annthyroid | 1.5053 | 0.9724 | 2.4778 |
>   | comm.and.crime | 1.0540 | 1.0901 | 2.1440 |
>   | imgseg | 0.4653 | 0.5770 | 1.0423 |
>   | satellite | 0.9737 | 0.8839 | 1.8576 |
>   | vertebral | 2.3124 | 0.6989 | 3.0113 |
>   | wine | 0.5662 | 1.0285 | 1.5946 |
>
>
> **q1 — Underperformance cases**
>
>   **Response:** Thank you for your insightful question. OFA-TAD does not achieve the best results on every dataset. Empirically, it underperforms most noticeably on very small-sample datasets (e.g., `glass` with 214 samples, `WPBC` with 198 samples), where context-set size limits the reliability of neighbor-distance estimation, and on high-dimensional datasets (e.g., `census` with ~500 features after encoding), where distance concentration under the curse of dimensionality may reduce the discriminability of neighbor-distance profiles. We will add this analysis to the revised paper.

---

> > ### Author Rebuttal · Reviewer_JEKV · 2026-04-02
> >
> > My concerns are fully addressed so I'd like to keep my score.

---

### Official Review · Reviewer_862v · 2026-03-13

**Soundness:** 3
**Presentation:** 3
**Significance:** 2
**Originality:** 3
**Overall Recommendation:** 3
**Confidence:** 4

**Summary:**

This paper attempts to address the Tabular Anomaly Detection (TAD) task under a “one model for all datasets” (OFA) setting by leveraging multiple source datasets. To tackle this problem, the authors propose a methodology that examines features through multiple views via various normalization transformations, and extracts anomaly-relevant information using a Mixture-of-Experts Scoring Network and Positional Embedding for assessing the importance of neighbor ranks. The proposed approach is empirically validated on a diverse collection of tabular datasets, demonstrating competitive performance.

**Compliance With Llm Reviewing Policy:**

Affirmed.

**Final Justification:**

Thank you for the clarifications. My concerns have been partially addressed, and I will increase my score accordingly.

**Key Questions For Authors:**

Weaknesses

In Section 3, the absence of explicit dimensionality annotations in the mathematical notation reduces readability. (Minor)
Upon examining Section 4.5 and Figure 5, it is observed that the gating weights assigned to the Raw and Quantile transformations are consistently low across most datasets. This raises the question of why these two transformations are included in the first place. Furthermore, there is a notable lack of ablation study on the selection of transformations themselves. (Major)
The rationale behind the selection of the four specific transformations is insufficiently justified. References [1, 2, 3] provide explanations for why neural networks tend to underperform tree-based models on tabular datasets (albeit in the context of classification and regression tasks). Grounding the choice of transformations in these insights — for instance, by connecting them to the spectral properties of tabular data — would substantially improve the quality and rigor of the paper. While the idea of viewing datasets through multiple transformation-induced perspectives is novel, stronger theoretical or empirical justification is needed. (Major)
Based on an examination of the datasets used, it appears that the benchmark datasets are sourced from [4]; however, this reference is not cited in the paper. (Minor)
As indicated in Appendix C, the proposed method incurs a relatively high computational complexity. However, there is no empirical comparison of inference time against baseline methods, making it difficult to assess the practical feasibility of the approach. (Minor)
While the Introduction frames the paper’s goal as solving the TAD task under the OFA setting, the content of Section 3 does not appear to address the OFA setting at all. In fact, reading Section 3 in isolation gives the impression that the paper is primarily concerned with extracting informative features under a conventional OFO setting. It is unclear how the methodology described in Section 3 is specifically connected to or motivated by the OFA objective. Furthermore, given that the proposed method aims to extract features that can be aligned across diverse domains, what concrete advantages does training on realistic multi-source datasets offer compared to using synthetically generated data (e.g., Gaussian-sampled data) as source datasets? Additionally, what is the performance when the source is reduced to a single dataset (e.g., training only on Wilt)? An analysis of this would help clarify the actual contribution of the multi-source training setup. (Major)
Sufficient details regarding the model architecture and experimental setup are missing. (e.g., optimizer, epoch) (Major)

Beyond the weaknesses listed above, the paper exhibits several additional shortcomings, including: the lack of robustness analysis for the selection of multi-source datasets used in training; the absence of a clear description of when and how all source data are mixed during training in Section 3; and a discrepancy in Figure 3, where the Metric Space depicted in the Multi-view Distance Encoding module appears — based on the main text — to be derived from a single dataset, yet is visually presented as if it originates from multiple distinct datasets. In light of these issues, I believe the paper in its current form is insufficient for acceptance at ICML.

Reference
[1] Shwartz-Ziv, Ravid, and Amitai Armon. “Tabular data: Deep learning is not all you need.” Information fusion 81 (2022): 84-90.
[2] Grinsztajn, Léo, Edouard Oyallon, and Gaël Varoquaux. “Why do tree-based models still outperform deep learning on typical tabular data?.” Advances in neural information processing systems 35 (2022): 507-520.
[3] Beyazit, Ege, et al. “An inductive bias for tabular deep learning.” Advances in Neural Information Processing Systems 36 (2023): 43108-43135.
[4] Han, Songqiao, et al. “Adbench: Anomaly detection benchmark.” Advances in neural information processing systems 35 (2022): 32142-32159.

**Limitations:**

See Weaknesses

**Strengths And Weaknesses:**

This paper addresses the TAD task under the OFA setting, which has been relatively unexplored in the community.
The authors conduct extensive experiments across a large number of tabular anomaly detection datasets.

---

> ### Author Rebuttal · Authors · 2026-03-31
>
> We are grateful to Reviewer 862v for recognizing that OFA-TAD addresses the relatively unexplored OFA setting in tabular anomaly detection, and for acknowledging the extensive experiments across a large number of datasets. Please find our detailed responses to your concerns below.
>
> **w1 — Notation / readability in Sec. 3**
>
> **Response:** We will add explicit dimensional annotations for key variables in Sec. 3.
>
> **w2&w3 — Multi-view transformation design and empirical justification**
>
> **Response:** A low average gating weight does not mean a view is useless—only less frequently dominant. View-level ablation (Table 1) shows that removing any view reduces performance, confirming complementarity. The four views are directly grounded in insights from [1-3]: `Raw` preserves original scale; `Std` and `MinMax` address scale/range sensitivity that drives NN underperformance on tabular data; `Quantile` handles the irregular distributions and function non-smoothness identified in [1-3]. MoE gating then adapts to the most informative view per dataset. We will add this discussion in the revised paper.
>
> Table 1. View-level ablation (AVG AUROC %)
> | Setting | AUROC |
> |:-|:-:|
> | **OFA-TAD** | **83.45±0.02** |
> | w/o quan | 82.55±0.05 |
> | w/o raw | 82.78±0.04 |
> | w/o minmax | 82.76±0.05 |
> | w/o std | 82.48±0.41 |
>
> **w4 — Missing ADBench citation**
>
> **Response:** We will add the ADBench citation and clarify the dataset source/split protocol.
>
> **w5 — Efficiency evidence missing**
>
> **Response:** We added unified efficiency comparison (Table 2). For OFO baselines, `total_time` includes both training (50 epochs) and testing. As shown in Table 2, OFA-TAD remains lightweight and is faster than several neural baselines, though slower than very light methods such as LUNAR.
>
> Table 2. Target-side efficiency comparison (mammography)
> | Method | Total time (s) |
> |:-|:-:|
> | AE | 8.74 |
> | DRL | 1.51 |
> | MCMTAD | 4.45 |
> | Disent-AD | 3.95 |
> | LUNAR | 0.20 |
> | OFA-TAD | 0.48 |
>
> **w6 — OFA-specificity unclear**
>
> **Response:** Unlike a typical OFO feature extractor, every component of Sec. 3 is designed to address a specific OFA challenge. Concretely: **Multi-View Distance Encoding**: Tabular datasets vary drastically in feature dimensionality and semantics across domains, making a shared raw-feature model infeasible. We design neighbor-distance profiles as a fixed-length, semantics-agnostic representation that is universally comparable across datasets. **Multi-View Transformation + MoE Gating**: Different datasets may reveal anomaly evidence in different induced metric spaces. We construct M complementary views and a gating module that assigns sample-specific weights, allowing the model to favor the most discriminative metric space for each target dataset without target-specific retraining.
>
> **w7 — Why realistic multi-source vs synthetic sources?**
>
> **Response:** Our goal is to learn anomaly cues that generalize under real cross-dataset shift. Because OFA-TAD relies on neighbor-distance profiles, source data must contain realistic local geometry, density variation, and anomaly separability—structures that simple synthetic distributions may fail to capture. Notably, we do use synthetic data in our pipeline: since real-world anomalies are scarce, we synthesize pseudo-anomaly samples to provide training supervision (Sec. 3.3). Using synthetically generated data as the source pretraining domain is a separate, promising direction we leave for future work.
>
> **w8 — Experimental details & Fig. 3 discrepancy**
>
> **Response:** OFA-TAD trains once on source datasets using Adam (15 epochs, lr=5e-4, wd=2e-5). Four fixed views, K=80, emb_dim=128, hidden_dim=64. Source datasets are used jointly but not concatenated—we construct multi-view distance features per dataset and optimize one shared model by alternating across epochs. Fig. 3's metric space is constructed per dataset/view, illustrating repeated pipeline application. During inference, target training split is used only for context retrieval/normalization.
>
> **w9 — Source robustness and reduced-source performance**
>
> **Response:** The dataset scaling experiments (Fig.7) in the paper have shown that when the source is reduced to a single dataset, the average performance is 81.93% AUROC and 64.45% AUPRC.  This experiment further shows consistent gains as more source datasets are added.
>
> To address source robustness more directly, we further constructed an additional fixed-target source-subset experiment: targets were fixed to `abalone|cover|wbc`, while we changed source triples across 3 subsets (`S1`-`S3`) and 5 seeds. Results are stable across subsets, indicating that the gain is not tied to one "lucky" source combination.
>
> **Table 3.** Source-subset robustness under fixed targets (mean+-std, %).
> | Set | AUROC | AUPRC |
> |:-:|:-:|:-:|
> | S1 | 90.92+-0.33 | 69.52+-3.14 |
> | S2 | 91.12+-0.21 | 71.05+-1.72 |
> | S3 | 91.39+-0.09 | 73.78+-1.42 |

---

> > ### Author Rebuttal · Reviewer_862v · 2026-04-04
> >
> > W1, W4, and W5 are considered resolved. However, the following concerns remain insufficiently addressed.
> >
> > W2/W3. A more detailed explanation of how each of the four views captures distinct anomaly evidence is still needed. We acknowledge that the remaining points have been adequately addressed.
> >
> > W6. The authors' explanation of how multi-source training is performed remains insufficiently detailed. More critically, we raise a concern regarding a potential failure mode: if the distance pattern of normal samples in Dataset A coincides with the distance pattern of anomalous samples in Dataset B, and both datasets are used as source datasets, the MoE network would receive contradictory training signals for the same input pattern. This could lead to a collapse in the learned scoring function.
> >
> > W7. We believe that empirical experiments directly comparing realistic multi-source data against synthetic sources are necessary to support this claim.
> >
> > W8/W9. While the authors provided some implementation details, several critical specifications remain missing, such as the number of layers in the MoE network, learning rate scheduler, and other architectural hyperparameters necessary for reproducibility.
> > Furthermore, while the authors have clarified in their rebuttal that the metric space in Figure 3 is constructed per dataset and view, the figure itself remains visually ambiguous and does not clearly convey this, potentially misleading readers. A revision of Figure 3 is therefore still needed.
> > Additionally, a more thorough analysis of how performance varies across different source combinations is required. While we acknowledge that exhaustive ablation over all possible source combinations is infeasible in a multi-source setting, understanding the robustness of the proposed framework requires experiments across a broader range of source combinations and target datasets.
> > For instance, randomly selecting 2 to 5 source datasets from ADBench and repeating this process approximately 30 times across diverse target datasets would provide a more convincing robustness analysis. Unlike vision or time-series domains where such experiments are computationally expensive, most datasets in ADBench are relatively small in terms of dimensionality and sample size, making such an analysis practically feasible.
> >
> > We thank the authors for their thoughtful responses. However, as the paper still requires substantial revisions across multiple aspects, we maintain our current score.

---

> > > ### Author Response · Authors · 2026-04-05
> > >
> > > We thank the reviewer for acknowledging that W1, W4, W5 are resolved. Our complete code has been uploaded for full reproducibility and serves as the authoritative reference for implementation specifics. Key details below.
> > >
> > > **W2/W3 — More detail on how each view captures distinct anomaly evidence**
> > >
> > > **Response:** [1-3] show that NNs underperform tree-based models on tabular data due to feature heterogeneity [3] and irregular target functions [2,3]. Our four views directly address these challenges by constructing complementary metric spaces: **Raw** preserves original scale, capturing absolute magnitude deviations; **Std** equalizes per-feature variance, addressing feature heterogeneity [3] so that relative deviations become visible; **MinMax** bounds features to [0,1] while preserving distribution shape, capturing boundary-region anomalies; **Quantile** rank-transforms to uniform, directly implementing the ranking-based smoothing advocated in [3] and mitigating the irregular functions identified in [2]. View-level ablation (Table 1, R1) confirms complementarity: removing any view degrades AUROC. MoE gating adaptively selects the most informative view per sample. We will expand this discussion in the revision.
> > >
> > > **W6 — Contradictory distance patterns across datasets**
> > >
> > > **Response:** This concern could potentially arise in raw feature space. For example, in network intrusion detection, high packet rates signal attacks, while in e-commerce high transaction volumes are normal business peaks. Such contradictions could indeed confuse a shared model. However, OFA-TAD operates in *distance-to-normal* space: all features are transformed into KNN distances to normal training samples only. The learning target is therefore universally consistent (*larger distance = more anomalous*), which is a direct consequence of the definition of anomaly, not a domain-specific heuristic. Normals are always close to normal neighbors and anomalies are always far, regardless of domain, so the contradictory scenario is unlikely to arise in practice. Our four views further apply different normalizations *before* distance computation, creating diverse neighborhood structures that reveal patterns invisible under any single view, and per-dataset quantile normalization maps all vectors to [0,1], ensuring numerically comparable inputs. Training iterates over source datasets per epoch with dataset-local feature extraction and no cross-dataset mixing (see uploaded code). All main experiments and the expanded robustness analysis (Table 8) confirm that the hypothesized concern does not manifest in practice.
> > >
> > > **W7 — Empirical comparison of realistic vs synthetic sources**
> > >
> > > **Response:** We agree this is a meaningful direction. However, designing a fair synthetic-source baseline is itself a non-trivial research problem, as it requires jointly specifying synthetic data distributions, dimensionality ranges, anomaly-to-normal ratios, and generation strategies. Each of these is an independent design axis whose choices could bias the comparison. A rushed or ad-hoc synthetic baseline risks producing misleading conclusions and would not meet the rigor expected at a top venue. We note that we do already leverage synthetic data within our pipeline: pseudo-anomaly generation (Sec. 3.3) synthesizes training supervision for each source dataset. Replacing *entire source datasets* with synthetic ones is a fundamentally different research question that we believe warrants a dedicated study rather than an auxiliary ablation within this paper.
> > >
> > > **W8/W9 — Architecture, Figure 3, and source-combination analysis**
> > >
> > > **Response:** We address each sub-point:
> > >
> > > **(a) Full architecture spec.** Input: (B,V,K,1). Per-view expert: Linear(1→128)+LayerNorm+pos_embed(K,128)→attention pooling→(B,128)→Linear(128→64→1)→per-view score. MoE gate: concat V embeddings (B,V×128)→Linear(V×128→64→V)→view weights. Final: Σ(scores×weights), MSE loss, no scheduler. Code uploaded for full reproducibility.
> > >
> > > **(b) Figure 3 revision.** We agree that Figure 3 could be made clearer and will revise it to explicitly annotate that the metric space is constructed per dataset and per view. We appreciate this constructive suggestion.
> > >
> > > **(c) Broader source-combination analysis.** Using the same 7 source and 34 target datasets in the paper, we randomly sampled source subsets of size 2–5, with 30 trials per size (exhaustive for sizes 2 and 5: C(7,2)=C(7,5)=21). Results in Table 8 show AUROC std below 0.17% across all sizes, with mean AUROC monotonically improving from 82.44% to 82.98%, confirming strong robustness to source selection.
> > >
> > > **Table 8.** Source-combination robustness (random trials, AVG AUROC %).
> > > | Source size | Valid trials | Mean AUROC | Std |
> > > |:-:|:-:|:-:|:-:|
> > > | 2 | 21 | 82.44 | 0.10 |
> > > | 3 | 30 | 82.74 | 0.17 |
> > > | 4 | 30 | 82.85 | 0.15 |
> > > | 5 | 21 | 82.98 | 0.15 |

---

### Decision · Program_Chairs · 2026-04-30

**Decision:**

Accept (regular)

**Comment:**

This paper proposes OFA-TAD, a one-for-all framework for tabular anomaly detection that trains once on multiple source datasets and transfers to unseen targets via multi-view neighbor-distance encoding and MoE-based anomaly scoring. Reviewers agreed that the problem is timely and practically relevant, and that the paper presents extensive experiments across a broad benchmark suite with generally strong results. The main concerns focused on fairness of comparison to OFO baselines, the justification and interpretation of the chosen transformations, inference/runtime costs due to multi-view kNN retrieval, and missing implementation or robustness details. The rebuttal addressed many of these issues by clarifying the OFA protocol, providing efficiency evidence, reporting multi-run results and additional source-robustness analyses, and supplying further implementation details. One reviewer remains unconvinced that the paper is fully mature, particularly regarding deeper justification of the multi-view design and broader robustness analysis, but the concerns do not point to a fundamental technical flaw. Overall, I view the paper as a technically solid and potentially useful contribution with moderate but meaningful novelty, and I recommend accept.